# The transcription factor odd-paired regulates temporal identity in transit-amplifying neural progenitors via an incoherent feed-forward loop

Merve Deniz Abdusselamoglu, Elif Eroglu[†], Thomas R Burkard, Jürgen A Knoblich*

IMBA – Institute of Molecular Biotechnology of the Austrian Academy of Science, Vienna Biocenter (VBC), Vienna, Austria

**Abstract** Neural progenitors undergo temporal patterning to generate diverse neurons in a chronological order. This process is well-studied in the developing *Drosophila* brain and conserved in mammals. During larval stages, intermediate neural progenitors (INPs) serially express Dichaete (D), grainyhead (Grh) and eyeless (Ey/Pax6), but how the transitions are regulated is not precisely understood. Here, we developed a method to isolate transcriptomes of INPs in their distinct temporal states to identify a complete set of temporal patterning factors. Our analysis identifies odd-paired (opa), as a key regulator of temporal patterning. Temporal patterning is initiated when the SWI/SNF complex component Osa induces D and its repressor Opa at the same time but with distinct kinetics. Then, high Opa levels repress D to allow Grh transcription and progress to the next temporal state. We propose that Osa and its target genes opa and D form an incoherent feedforward loop (FFL) and a new mechanism allowing the successive expression of temporal identities.
DOI: https://doi.org/10.7554/eLife.46566.001

*For correspondence: juergen.knoblich@imba.oeaw.ac.at

Present address: [†]Department of Cell and Molecular Biology (CMB), Karolinska Institutet, Stockholm, Sweden

**Competing interests:** The authors declare that no competing interests exist.

## Introduction

During brain development, neural stem cells (NSCs) generate large numbers of highly diverse neuronal and glial cells in chronological order (*Cepko et al., 1996*; *Gao et al., 2014*; *Greig et al., 2013*; *Holguera and Desplan, 2018*). Through a phenomenon known as temporal patterning, NSCs acquire properties that change the fate of their progeny over time (*Kohwi et al., 2013*; *Mattar et al., 2015*; *Okamoto et al., 2016*). Importantly, temporal patterning of NSCs is an evolutionary conserved process and has been observed across species ranging from insects to mammals (*Alsiö et al., 2013*; *Livesey and Cepko, 2001*; *Toma et al., 2014*). During mammalian brain development, neural progenitors in the central nervous system (CNS) undergo temporal patterning by relying on both extrinsic as well as progenitor-intrinsic cues. Wnt7, for example, is an extracellular ligand required for the switch from early to late neurogenesis in cortical progenitors (*Wang et al., 2016*), Ikaros (the ortholog of the Drosophila Hunchback), in contrast, is an intrinsic factor specifying early-born neuronal fates (*Mattar et al., 2015*). Like Ikaros, intrinsic temporal identity factors in vertebrates are often homologous to factors described in *Drosophila* (*Naka et al., 2008*; *Ren et al., 2017*; *Syed et al., 2017*). How these factors are involved in neuronal fate specification and how they are regulated remain unknown.

*Drosophila* has been crucial to understanding stem cell biological mechanisms and in particular distinct temporal patterning processes (*Homem and Knoblich, 2012*). During embryonic neurogenesis, *Drosophila* NSCs, called Neuroblasts (NBs), undergo temporal patterning through a cascade of

**eLife digest** The brain consists of billions of neurons that come in a range of shapes and sizes, with different types of neurons specialized to perform different tasks. Despite their diversity, all of these neurons originate from a single population known as neural stem cells. As the brain develops, each neural stem cell divides to produce two daughter cells: one remains a stem cell, which can then divide again, and the other becomes a neuron.

A longstanding question in developmental biology is how a limited pool of neural stem cells can generate so many different types of neurons. The answer seems to lie in a process known as temporal identity, whereby neural stem cells of different ages give rise to different types of neurons. This requires neural stem cells to keep track of their own age, but it is still unclear how they can do so.

Abdusselamoglu et al. have now uncovered part of the underlying mechanism behind temporal identity by studying fruit flies, an insect in which the early stages of brain development are similar to the ones in mammals. A method was developed to sort fly neural stem cells into groups based on their age. Comparing these groups revealed that a protein called Opa make neural stem cells switch from being 'young' to being 'middle-aged'. Another protein, Osa activates Opa, which in turn represses a protein called Dichaete. As Dichaete is mainly active in young neural stem cells, the actions of Osa and Opa push neural stem cells into middle age.

Fruit flies are therefore a valuable system with which to study the mechanisms that regulate neural stem cell aging. Revealing how the brain generates different types of neurons could help us study the way these cells organize themselves into complex circuits. This knowledge could then be harnessed to understand how these processes go wrong and disrupt development.

DOI: https://doi.org/10.7554/eLife.46566.002

transcription factors (*Isshiki et al., 2001*). During larval neurogenesis, NB temporal patterning relies on opposing gradients of two RNA-binding proteins (*Liu et al., 2015*; *Syed et al., 2017*). Temporal patterning is also seen in intermediate neural progenitors (INPs), the transit-amplifying progeny of a discrete subset of larval NBs called type II NBs (*Bayraktar and Doe, 2013*). Once they arise from an asymmetric division of a type II NB, newborn INPs undergo several maturation steps before they resume proliferation: they first turn on earmuff (erm), and Asense (ase), and finally Deadpan (Dpn) expression to become mature INPs (mINP) (*Bello et al., 2008*; *Boone and Doe, 2008*; *Bowman et al., 2008*; *Janssens et al., 2014*; *Walsh and Doe, 2017*). Then mINPs divide 3–6 times asymmetrically to generate ganglion mother cells (GMCs), which in turn divide to generate a pair of neurons or glia. Analogous to embryonic NBs (*Isshiki et al., 2001*), recent reports suggest that a transcription factor cascade regulates temporal patterning of INPs (*Bayraktar and Doe, 2013*). Indeed, the sequential expression of Dichaete (D), Grainhead (Grh) and Eyeless (Ey) is required to generate different neurons: D[+] INPs produce Brain-specific homeobox (Bsh)[+] neurons, while Ey[+] INPs produce Toy[+] neurons (*Bayraktar and Doe, 2013*).

The three temporal identity factors are regulated through various regulatory interactions (*Bayraktar and Doe, 2013*; *Doe, 2017*): D is necessary, but not sufficient, for activating Grh. Grh instead is required for repression of D and activation of Ey (*Bayraktar and Doe, 2013*). Therefore, INP temporal patterning is thought to be regulated by a 'feedforward activation and feedback repression' mechanism (*Figure 1A*). Intriguingly however, INP temporal patterning also critically requires the SWI/SNF chromatin remodeling complex subunit Osa (*Eroglu et al., 2014*). Although Osa is not considered a specific temporal identity factor, it is required to initiate temporal patterning by activating the initial factor D. While the Osa target gene hamlet is required for the Grh-to-Ey transition (*Eroglu et al., 2014*), regulation of the first transition is less well understood. This result suggests that in addition to feedforward activation and feedback repression, temporal switch genes are required to ensure correct INP temporal patterning. Nevertheless, D and ham double knock down (k.d.) phenotypes do not recapitulate the complete loss of temporal patterning initiation observed in Osa-depleted type II NB lineages, suggesting the contribution of additional unidentified factors.

Here, we describe a FACS-based method to isolate INPs from three different temporal identities. By comparing the transcriptomic profiles of each set of INPs, we identify odd-paired (opa), a

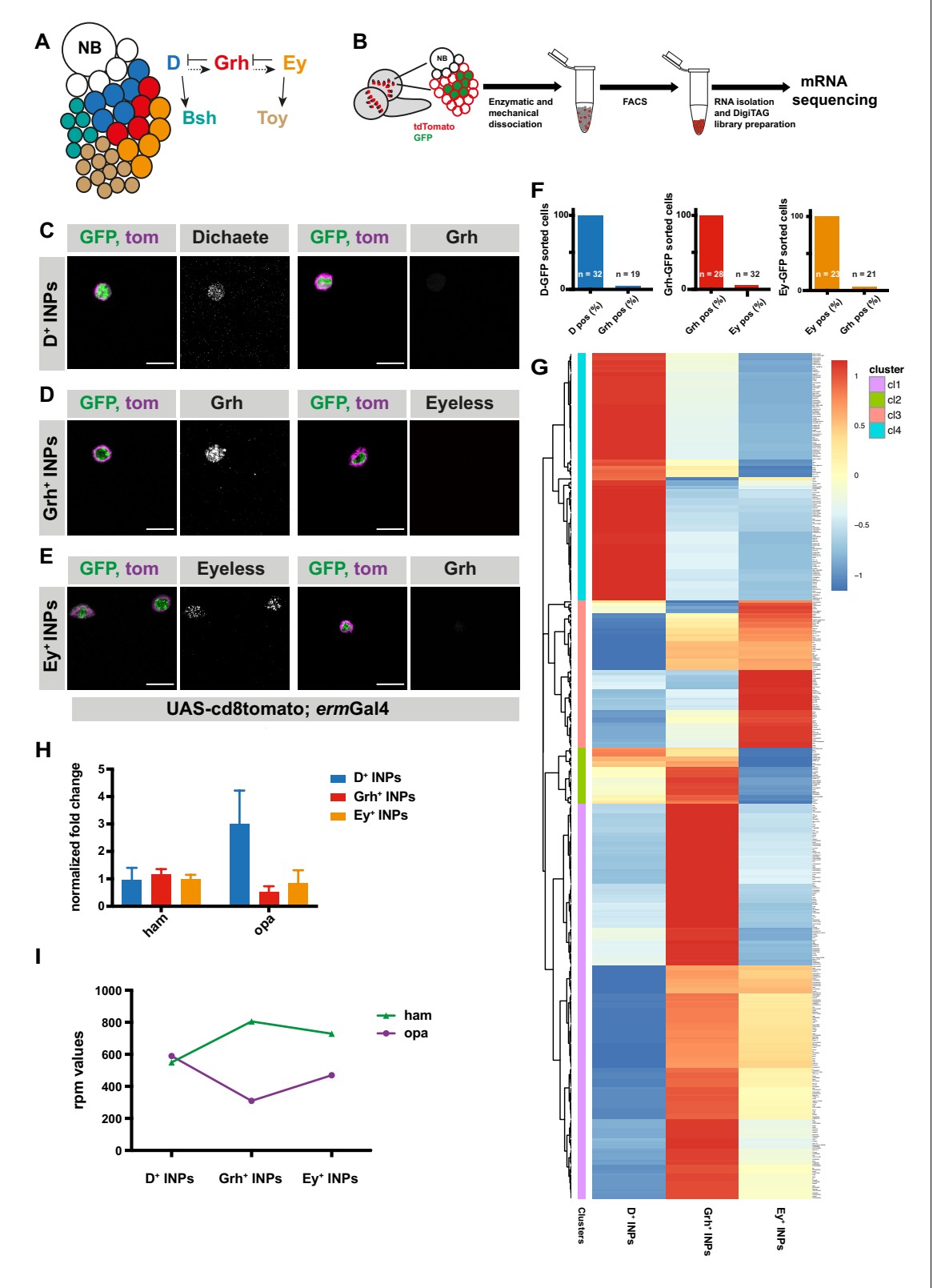

**Figure 1.** Transcriptomic analysis of temporally staged-INPs. (**A**) Cartoon depicting a typical type II neuroblast of larval *Drosophila* brain; NB and imINPs (empty circles) are followed by mINPs and neurons, GMCs omitted for simplicity. INPs are temporally patterned with Dichaete (blue), Grainyhead (red), and Eyeless (orange), and neurons are Bsh (green) or Toy (brown) positive. Summary of the regulation of temporal identity factors, and their progeny. (**B**) Cartoon illustrating the strategy used to isolate temporally-staged INPs. (**C–E**) D-, Grh and Ey-GFP FACS-sorted cells are stained

*Figure 1 continued on next page*

*Figure 1 continued*

for D and Grh (**C**), Grh or Ey (**D–E**), GFP-tagging temporal identity factors (in green, D or, Grh or Ey), tdTomato tagging the membrane of INPs (magenta), antibody staining (gray) scale bar 10 μm, (induced with ermGal4, marked with membrane bound tdTomato). (**F**) Graphs showing the percentage of temporal identity positive cells in D-, Grh- or Ey-GFP FACS sorted cells. n numbers are depicted on the graphs. (**G**) Hierarchical clustering analysis of gene log2fc between three different temporally-staged INP populations. (**H**) qPCR analysis of opa and ham expression levels in FACS-sorted D$^+$, Grh$^+$ and Ey$^+$ INPs. Data are mean ± SD, n = 3, genes were normalized to Act5c, and then the average expression levels, Delta-Delta Ct method is used. (**I**) Graph showing the rpm levels of opa and ham between different INP temporal stages.

DOI: https://doi.org/10.7554/eLife.46566.003

The following source data and figure supplements are available for figure 1:

**Source data 1.** Quantification of temporally FACS-sorted INPs for temporal markers (*Figure 1F*).

DOI: https://doi.org/10.7554/eLife.46566.006

**Source data 2.** qPCR data (*Figure 1H*).

DOI: https://doi.org/10.7554/eLife.46566.007

**Source data 3.** Rpm levels of opa and ham genes in three different temporal states of INPs (*Figure 1I*).

DOI: https://doi.org/10.7554/eLife.46566.008

**Figure supplement 1.** INPs can be FACS-sorted depending on their temporal identity.

DOI: https://doi.org/10.7554/eLife.46566.004

**Figure supplement 1—source data 1.** Quantification of number of INPs in three different temporal states versus their GFP-tagged counterparts (*Figure 1—figure supplement 1A*).

DOI: https://doi.org/10.7554/eLife.46566.009

**Figure supplement 1—source data 2.** Quantification of FACS-sorted INPs Dpn staining positivity (*Figure 1—figure supplement 1D*).

DOI: https://doi.org/10.7554/eLife.46566.010

**Figure supplement 1—source data 3.** qPCR data (*Figure 1—figure supplement 1E*).

DOI: https://doi.org/10.7554/eLife.46566.011

**Figure supplement 1—source data 4.** Rpm levels of genes in three different temporal states of INPs (*Figure 1—figure supplement 1F*).

DOI: https://doi.org/10.7554/eLife.46566.012

**Figure supplement 2.** Temporally sorted INPs are pure populations.

DOI: https://doi.org/10.7554/eLife.46566.005

transcription factor whose expression is enabled by direct binding of Osa to its TSS, as a regulator of temporal patterning and repressor of D. Though Osa enables both D and Opa expression, Opa's slower activation kinetics allow D to function in a short time window before being repressed by Opa. This mode of action resembles an incoherent feedforward-loop (FFL) motif, where an upstream gene directly activates the target gene, meanwhile indirectly repressing it by activating its repressor (*Alon, 2007*; *Mangan and Alon, 2003*). Thus, we uncover a novel mechanism controlling temporal patterning during neurogenesis.

## Results

### Transcriptome analysis of distinct INP temporal states

To obtain a comprehensive list of temporally regulated genes in INPs, we used FACS to purify INPs at each of their three temporal states: D$^+$, Grh$^+$ and Ey$^+$ (*Figure 1B*). For this, we generated fly lines expressing tdTomato under an INP specific promoter (erm-Gal4 >CD8::tdTomato) and expressing GFP-fusions of one of the temporal identity factors (D-GFP, Grh-GFP and Ey-GFP, *Figure 1—figure supplement 1A*). Although D-GFP flies were generated with CRISPR/Cas9 method to knock-in GFP into the endogenous locus, Grh-GFP and Ey-GFP flies were generated as BAC clones insertions (*Spokony and White, 2012*). To test if extra copies from BAC clones cause overexpression effects, numbers of each temporal state were quantified in control versus GFP-tagged brains (*Figure 1—figure supplement 1A*). After dissection and dissociation of third instar larval brains, GFP-positive INP populations (D-GFP$^+$, Grh-GFP$^+$ and Ey-GFP$^+$) were identified (*Figure 1B* and *Figure 1—figure supplement 1B*) as the largest cells with highest GFP and tdTomato expression (*Figure 1—figure supplement 1B*). Using immunofluorescence (IF), these cells were verified to be mature INPs (*Figure 1—figure supplement 1C-D*). All sorted cells within the INP populations expressed Dpn, indicating a 100% mature INP identity, while unsorted cells showed a mixture of Dpn$^+$ and Dpn$^-$ cells (*Figure 1—figure supplement 1C-D*). We validated the temporal identity of the progenitors by

performing IF for their respective temporal identity markers (*Figure 1C–F* and *Figure 1—figure supplement 2*). Importantly, each GFP$^+$ sorted INP population was 100% positive for its respective temporal marker (*Figure 1F*). In contrast, the unsorted cells consisted of mixed cell populations containing various temporal identities (*Figure 1—figure supplement 2B*). Lastly, we tested for the presence of sorted cells expressing markers of two temporal identities, which reflects transition states of INP temporal patterning as occurs in vivo. Analyzing Grh IF on D-GFP$^+$ and Ey-GFP$^+$ sorted cells, and Ey IF on Grh-GFP$^+$ sorted cells revealed that sorted populations contained only 4–6% of such double-positive cells (*Figure 1C–F*, and *Figure 1—figure supplement 2A-C*), suggesting we can isolate almost pure populations of different temporal states. Collectively, we established the genetic tools and methodology to precisely sort INPs into separate populations according to their three distinct temporal states.

Since our stringent FACS sorting conditions led to low RNA yields, we generated cDNA libraries using DigiTag (*Landskron et al., 2018*; *Wissel et al., 2018*). With this RNA sequencing strategy, we found 458 genes expressed differently between D$^+$ and Grh$^+$ INPs, and 466 genes between Grh$^+$ and Ey$^+$ INPs (FDR 0.05, log2foldchange > 1, and Rpm (reads per million mapped reads)>10 in one of three samples/D$^+$, Grh$^+$ or Ey$^+$ INPs). Hierarchical clustering identified genes specifically expressed in certain temporal states, and therefore potentially involved in temporal patterning (*Figure 1G*). First, we confirmed the quality of our dataset by examining the transcriptional changes of temporal identity genes with quantitative PCR (qPCR) (*Figure 1—figure supplement 1E*). As expected, each temporal state had high expression levels of their own temporal identity genes. Second, we confirmed the expression of known temporal identity genes (*Figure 1—figure supplement 1F*). FACS-purified Grh$^+$ INPs expressed high levels of Ey mRNA. However, immunofluorescent analysis showed that Grh$^+$ INPs expressed only low levels of Ey protein, suggesting that post-transcriptional modifications regulate the Grh-to-Ey transition (*Figure 1C–F* and *Figure 1—figure supplement 1F*). Third, we performed GO-term analysis on the identified gene clusters. Genes upregulated in D$^+$ INPs showed enrichment for mitochondrial translation, cellular nitrogen compound metabolic process and gene expression (*Figure 1—figure supplement 2D*). Genes upregulated in Grh +INPs were enriched for protein binding and system development (*Figure 1—figure supplement 2E*). Finally, genes upregulated in Ey +INPs were enriched for neurogenesis and sequence-specific DNA binding (*Figure 1—figure supplement 2F*). Interestingly, we observed that the glial identity-promoting factor glial cell missing (gcm) and cell cycle inhibitor dacapo (dap) were upregulated in Ey$^+$ INPs (*Figure 1G—figure supplement 1F*). These observations support previous findings indicating that INPs begin producing glia cells instead of neurons during their later cell divisions, and that Ey is required for cell cycle exit (*Baumgardt et al., 2014*; *Bayraktar and Doe, 2013*; *Ren et al., 2018*; *Viktorin et al., 2013*). To identify genes that regulate transitions of temporal patterning, we focused on genes with a dynamic expression pattern between INP populations. To this end, we focused on genes with a log2foldchange > 1 in either the D-to-Grh or Grh-to-Ey transition. From this list, we excluded genes with a log2foldchange < 0.5 in the remaining transition. We applied a cut-off of Rpm (reads per million mapped reads)>50 in one of the three temporal identity states due to the fact that all the other temporal identity factors, along with osa and ham, had high expression levels. With these criteria, we identified 71 genes (*Supplementary file 1* and *Supplementary file 2*), 49 of which displayed an expression pattern of high in D + INPs, low in Grh +INPs, and finally higher in Ey +INPs. Among these genes, odd-paired (opa) was ranked as the 5$^{th}$ hit that is most downregulated in Grh$^+$ INPs (*Figure 1G–I*, *Supplementary file 1*). Since Osa binds to the TSS of opa in order to prime its expression (*Eroglu et al., 2014*), we investigated in detail the potential role of Opa in regulating INP temporal patterning.

## Odd-paired (opa) is required for the progression of INP temporal patterning

Opa is a transcription factor containing five zinc finger domains and is essential for para-segmental subdivision of *Drosophila* embryos (*Benedyk et al., 1994*; *Mizugishi et al., 2001*). During development, Opa ensures the timely activation of the transcription factors engrailed and wingless (*Benedyk et al., 1994*). To test if opa regulates INP temporal patterning, we depleted opa using RNAi expressed specifically in INPs with ermGal4. Opa knockdown slightly increased the total number of INPs (Dpn$^+$ cells), but drastically increased the number of D$^+$ INPs while decreasing the number of both Grh$^+$ and Ey$^+$ INPs (*Figure 2A–D*). We confirmed this result by performing mosaic

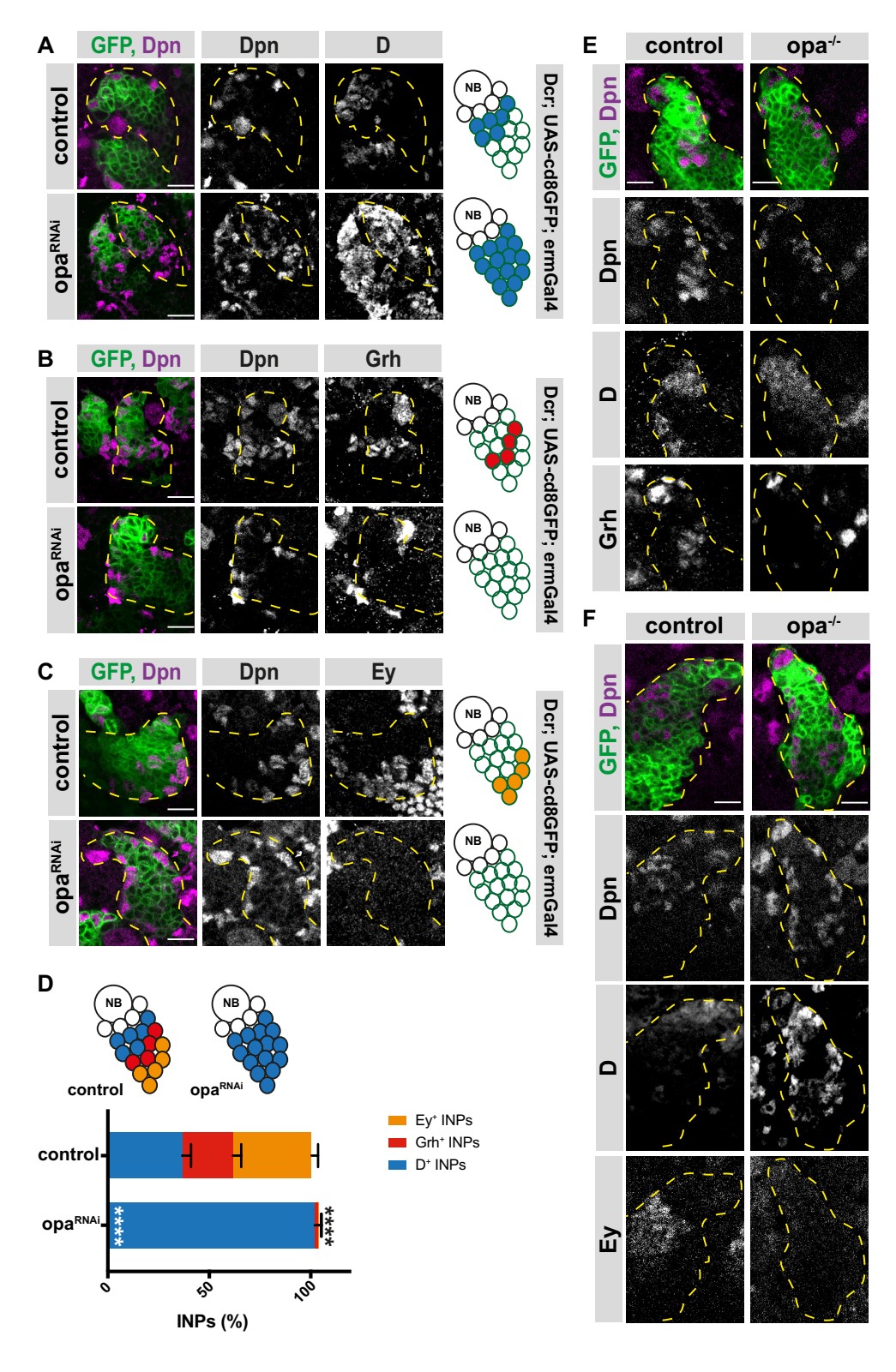

**Figure 2.** Opa is required for the progression of temporal patterning of INPs. (A) Close-up images of larval brains expressing RNAi against opa in INPs, stained for Dpn and D (induced with ermGal4, marked with membrane bound GFP). Lineages are outlined with yellow dashed line. (B) Close-up images of larval brains expressing RNAi against opa in INPs, stained for Dpn and Grh (induced with ermGal4, marked with membrane bound GFP). Lineages are outlined with yellow dashed line. (C) Close-up images of larval brains expressing RNAi against opa in INPs, stained for Dpn and Ey (induced with

*Figure 2 continued on next page*

*Figure 2 continued*

ermGal4, marked with membrane bound GFP). Lineages are outlined with yellow dashed line. (**D**) Quantification of INP numbers in different temporal stages identified by antibody staining of $Dpn^+$, $D^+$ cells, $Dpn^+$, $Grh^+$ cells, and $Dpn^+$, $Ey^+$ cells in control and opa knock-down brains, n = 10, total INP numbers in control were normalized to 100%. Data represent mean ± SD, ***p<=0.001, Student's t-test ($D^+$ INPs control 12.44 ± 1.42 [n = 10], opa RNAi 34.66 ± 1.02 [n = 12], p<0.001; $Grh^+$ INPs control 8.5 ± 1.32 [n = 10], opa RNAi 0.5 ± 0.65 [n = 12], p<0.001; $Ey^+$ INPs control 13.2 ± 0.98 [n = 10], opa RNAi 0.2 ± 0.4 [n = 10], p<0.001). (**E**) Control and opa mutant MARCM clones marked by membrane-bound GFP, stained for Dpn, Grh and D after 120 hr of induction. Control clone has $D^+$, $Dpn^+$ INPs followed by $Grh^+$ INPs while opa mutant clone has increased number of $D^+$ INPs and decreased number of $Grh^+$ INPs. (**F**) Control and opa mutant MARCM clones marked by membrane-bound GFP, stained for Dpn, D and Ey after 120 hr of induction. Opa mutant clone has higher number of $D^+$ INPs and lower number of $Ey^+$ INPs. Scale bar 10 μm in all images.
DOI: https://doi.org/10.7554/eLife.46566.013

The following source data and figure supplements are available for figure 2:

**Source data 1.** Quantification of number of INPs in three different temporal identities between control versus opa-depleted brains with INP-specific driver (*Figure 2D*).
DOI: https://doi.org/10.7554/eLife.46566.016

**Figure supplement 1.** Opa is required for D repression.
DOI: https://doi.org/10.7554/eLife.46566.014

**Figure supplement 1—source data 1.** Quantification of number of INPs in three different temporal identities between control versus opa-depleted brains with INP-specific driver in DM1 lineages (*Figure 2—figure supplement 1C*).
DOI: https://doi.org/10.7554/eLife.46566.017

**Figure supplement 2.** Opa regulates the transition from D-to-grh.
DOI: https://doi.org/10.7554/eLife.46566.015

**Figure supplement 2—source data 2.** Quantification of number of INPs in three different temporal identities between control versus opa-depleted brains with type II-specific driver (*Figure 2—figure supplement 2D*).
DOI: https://doi.org/10.7554/eLife.46566.018

analysis with a repressible cell marker (MARCM) to create mosaic opa (-/-) mutant or control opa (+/+) GFP$^+$ cell clones (*Lee and Luo, 1999*). Control clones were indistinguishable from WT, whereas opa mutant clones contained predominantly $D^+$ INPs, at the expense of the other two temporal states (*Figure 2E–F*). The RNAi and mosaic mutant analysis both indicate that loss of Opa causes a shift in INP temporal state identity such that the early generated $D^+$ INPs are increased while the later generated $Grh^+$ and $Ey^+$ INPs are decreased. These results suggest that opa is regulating the D-to-Grh transition by either repressing D or activating Grh. Since it has been previously shown that Grh is not sufficient for D repression (*Bayraktar and Doe, 2013*), we tested whether the main role of opa is to repress D. For this, we depleted opa in DM1 lineages, which undergo temporal patterning by expressing only D and then Ey (*Figure 2—figure supplement 1*). Opa knock-down in DM1 lineages caused a significant increase in the number of $D^+$ INPs at the expense of $Ey^+$ INPs, suggesting that opa is required for D repression (*Figure 2—figure supplement 1*).

Finally, we tested if opa regulates processes upstream of temporal patterning during the stages of initial INP maturation with a type II-specific driver line. When expressing opa RNAi specifically in type II NBs, we observed no effect on INP maturation (*Figure 2—figure supplement 2A*) as observed with sequential activation of Ase and Dpn, but immunofluorescent analysis of INPs for D, Grh and Ey expression showed the same phenotype as INPs depleted for opa (*Figure 2—figure supplement 2B-D*). Collectively, these data suggest that opa inhibits D expression. Furthermore, similar to hamlet, Opa appears to act as a temporal identity switch gene, controlling the transition from a $D^+$ to a $Grh^+$ state. To test if opa knock-down impairs INP asymmetric cell division leading to the disruption in temporal patterning, we analyzed the expression of Mira, a known scaffolding protein that localizes asymmetrically during cell division, and aPKC, which localizes to apical cortex (*Figure 2—figure supplement 2E*). Opa-depleted INPs can asymmetrically segregate Mira and aPKC, which suggests that asymmetric division is normal. Thus, opa is indeed a temporal switch factor required for the D-to-Grh state.

## Opa regulates the transition from early to late born neurons and is required for motor function

INP temporal patterning results in the production of different neuronal subtypes at distinct periods of neurogenesis. For instance, 'young', $D^+$ INPs produce Brain-specific homeobox (Bsh)$^+$ neurons

and 'old', Ey$^+$ INPs produce Toy$^+$ neurons (**Bayraktar and Doe, 2013**). Since the progression of INP temporal identity is disrupted in opa-depleted INPs, we tested whether this disrupted identity affects the production of different types of neurons. INP-driven opa RNAi displayed a significant increase in Bsh$^+$ neurons, at the expense of Toy$^+$ neurons (**Figure 3A–C**). In addition, opa-depleted MARCM clones also contained increased numbers of Bsh$^+$ neurons compared to wild-type counterparts (**Figure 3D**). This result confirms that shifting the INP identity toward a D$^+$ identity leads to a concomitant increase in the Bsh$^+$ neurons produced by D$^+$ INPs. Thus, altering the temporal identity progression of neural progenitors can alter the proportions of neuronal subtypes in the brain.

We next investigated whether altering the proportions of neuronal subtypes leads to a defect on brain morphology and function. The adult central complex (CCX) brain region relies on type II NB neurogenesis (**Bayraktar et al., 2010**; **Izergina et al., 2009**). Opa-depletion in INPs caused major alterations in the gross morphology of the adult CCX. The fan-shaped body (FB) was enlarged, the noduli (NO) and ellipsoid body (EB) only partially formed, and the protocerebral bridge (PB) appeared fragmented (**Figure 3E**). Since the CCX is required for adult motor functions (**Callaerts et al., 2001**; **Young and Armstrong, 2010**), we tested whether altered CCX morphology affected motor behavior. Compared to control flies, INP-driven opa RNAi caused impaired negative geotaxis performance (**Figure 3F**). Thus, opa is a temporal switch gene required for neuronal subtype specification, which is required for the correct assembly and function of the adult central complex. Thus, the temporal identity specification of neural progenitors is crucial for proper neural cell complexity, and brain function.

## Dichaete and Opa are sequentially expressed in INPs

If opa is required for the D-to-grh transition, what is the molecular mechanism of this transitional regulation? To answer this question, we first confirmed that opa is indeed a target of Osa in type II NB lineages by analyzing opa protein expression within the NB lineage, and whether this expression is regulated by Osa. We generated healthy, homozygous, endogenously C-terminally tagged opa::V5 knock-in flies (**Figure 4—figure supplement 1A**). Through immunofluorescent analysis of V5 tag expression, we observed that Opa is expressed throughout the type II lineage in INPs (marked with Dpn and Ase) and, GMCs (Pros$^+$ cells) and neurons, but not in NBs (Dpn$^+$) or immature INPs (Dpn$^-$/Ase$^-$ or Dpn$^-$/Ase$^+$ cells) (**Figure 4—figure supplement 1B-D**). Opa is also expressed in the DM1 lineage, even though DM1 lineages display a temporal patterning lacking Grh expression (**Figure 1—figure supplement 1E**). To check the specificity of the opa-V5 line, we depleted opa specifically in type II lineages using RNAi. As expected, opa-V5 expression decreased with opa-RNAi (**Figure 4—figure supplement 1E-F**). The proper expression of opa is dependent on Osa, since Osa-knockdown in type II NBs resulted in a loss of Opa (**Figure 4—figure supplement 2A and B**).

Since both D and opa are direct Osa targets, we next compared the expression pattern of D and opa (**Figure 4A**). Without exception, D$^+$/opa$^-$ INPs appeared before D$^+$/opa$^+$ cells in the lineage (**Figure 4A**). However, in later temporal states, all Grh$^+$ and Ey$^+$ INPs expressed opa (**Figure 4B**, and **Figure 4—figure supplement 3A**). Our transcriptome data suggest that opa expression fluctuates throughout the three different INP populations. To confirm this hypothesis, we calculated the intensity of the opa-V5 signal among these three populations (**Figure 4C–D**, and **Figure 4—figure supplement 3B**). Indeed, we found that D$^+$ INPs express the highest opa protein levels (**Figure 4C**), while Grh$^+$ INPs express the lowest (**Figure 4D** and **Figure 4—figure supplement 3B**). Since D expression precedes opa expression, it is possible that D activates opa. However, upon type II NB specific D knockdown, opa localization was unchanged (**Figure 4E**). Interestingly, D knockdown alone also did not prevent later temporal stages, Grh and Ey, to appear (**Bayraktar and Doe, 2013**), suggesting that other factor(s) are required to maintain temporal identities in INPs. Since Osa-depleted type II NB lineages fail to initiate temporal patterning (**Eroglu et al., 2014**), we hypothesized that one of these unidentified factors could be a target of Osa that remains expressed in D-depleted INPs, such as opa. To test this hypothesis, we examined the epistatic genetic interactions between D and Opa. Double knock down of D and opa by type II NB-specific RNAi produced type II lineages containing fewer Dpn$^+$/Ase$^+$ INPs compared to controls (**Figure 4F–G**). This result suggests that even though D and opa are Osa targets, two of them alone cannot fully account for Osa tumor suppressor role (**Figure 4F–G**). Importantly, all known temporal identity markers on the remaining cells were absent, suggesting a complete loss of temporal identity in these INPs (**Figure 4F–G**). However, since these cells also lost their INP identity due to lack of Dpn and Ase,

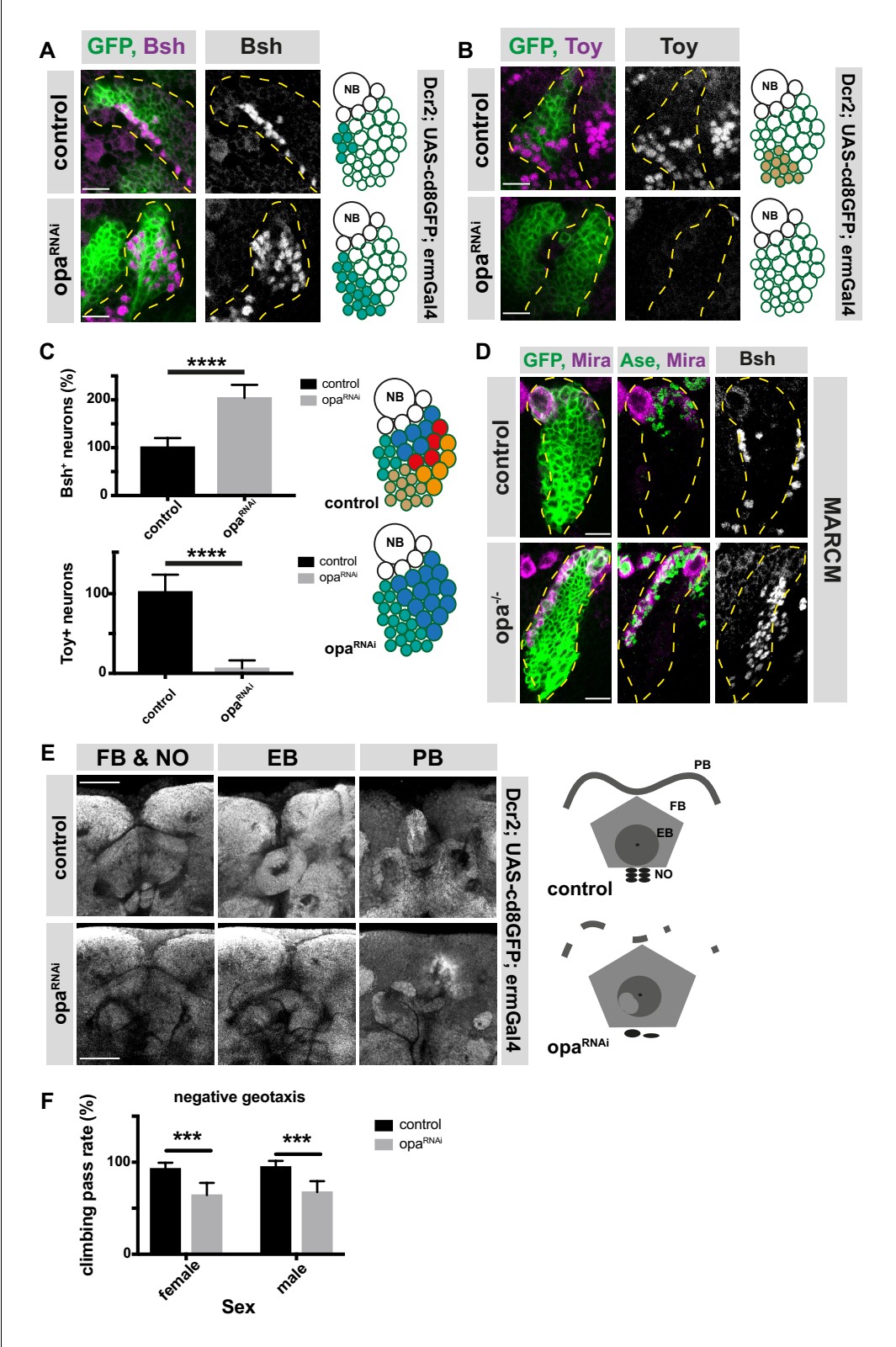

**Figure 3.** opa is an important factor for the generation of both early and late-born INP progeny and contributes to adult brain central complex. (A–B) Close-up images of larval brains expressing RNAi against opa in INPs, immunofluorescence for Bsh (A), and Toy (B) neuronal markers, scale bar 10 µm, lineages are outlined with yellow dashed line (induced with ermGal4, marked with membrane bound GFP). (C) Quantification of Bsh+ and Toy+ neurons in control and opa knock-down brains, n = 11, total Bsh+ or Toy+ neuron numbers in control were normalized to 100%. Data represent mean ± SD,

*Figure 3 continued on next page*

*Figure 3 continued*

***p<=0.001, Student's t-test. (D) Control and opa mutant MARCM clone marked by membrane-bound GFP, stained with Mira, Ase, and Bsh antibodies after 120 hr of induction. The clones are marked with yellow dashed line, scale bar 10 μm. (E) Close-up images of adult central complex, composed of fan-shaped body (FB), noduli (NO), ellipsoid body (EB), and protocerebral bridge (PB) of control and opa knock-down brains, stained with Bruchpilot antibody (gray) (induced with ermGal4) scale bar 50 μm. (F) Negative geotaxis assay with control and opa RNAi expressing flies (induced with ermGal4, marked with membrane bound GFP). For each genotype n = 10 replicates, each consisting of 10 adult female or male adults. Data are mean ± SD, ***p<0.001, Student's t-test.

DOI: https://doi.org/10.7554/eLife.46566.019

The following source data is available for figure 3:

**Source data 1.** Quantification of Bsh$^+$ or Toy$^+$ neuron numbers in control versus opa-depleted brains with INP-specific driver (*Figure 3C*).

DOI: https://doi.org/10.7554/eLife.46566.020

**Source data 2.** Quantification of the percentage pass rate of flies with control versus opa-depleted brains (*Figure 3F*).

DOI: https://doi.org/10.7554/eLife.46566.021

they exhibit a different phenotype than Osa knockdown. Therefore, our data suggest that opa is required for the repression of D, the activation of Grh, and thus the progression of temporal identities in INPs.

## Opa is an expression level-dependent repressor of D

If Opa suppresses D, one puzzling aspect of our data is the presence of double-positive D$^+$/opa$^+$ INPs (*Figure 4A*). To better understand this paradox, we overexpressed opa in type II NBs during a period before D is normally expressed. Overexpression of opa resulted in shorter lineages (*Figure 5—figure supplement 1A-B*), decreased total INP numbers (*Figure 5—figure supplement 1A*), and a loss of type II NBs (marked by Dpn or Mira) (*Figure 5—figure supplement 1A-B*). Co-expressing the apoptosis inhibitor p35 did not prevent NB loss or shortened lineages, suggesting that opa overexpression does not induce cell death, but causes premature differentiation instead (*Figure 5—figure supplement 1C*). NBs and INPs overexpressing opa successfully segregated Mira and aPKC, excluding that asymmetric cell division was altered (*Figure 5—figure supplement 1D-E*, and *Figure 2—figure supplement 2E*). Overexpressing opa in type II NB lineages caused complete loss of D$^+$ INPs, but the few remaining INPs could still activate Grh and Ey (*Figure 5A–C*), which is similar to D knockdown phenotype (*Bayraktar and Doe, 2013*).

To exclude that these could result from altered NB patterning, we next overexpressed opa in an INP-specific manner during a stage where D is normally expressed. Opa overexpression caused a decrease in D$^+$ INPs (*Figure 5D–F*), and a concomitant increase in both Grh$^+$ and Ey$^+$ INP populations (*Figure 5D–F*). This result further indicates that Opa represses the early D$^+$ temporal identity, but also activates later Grh$^+$ temporal identity. We also overexpressed opa in DM1 lineages in an INP-specific manner, which resulted in a decrease in D$^+$ INP numbers and an increase in Ey$^+$ INPs (*Figure 5—figure supplement 2A and C*). However, ectopic Grh expression was undetectable (*Figure 5—figure supplement 2B*), suggesting opa mis-expression does not cause ectopic Grh expression. Collectively, these results show that opa-mediated repression of D depends on Opa expression levels.

## Opa and ham together control the correct representation of each temporal identity

Having established an interaction between opa and D, we next wondered if opa and ham, two temporal switch genes, can recapitulate the Osa loss-of-function phenotype, a more upstream regulator of lineage progression in type II NBs. Osa knock-down causes INPs to revert back to the NB-state due to a failure to initiate temporal patterning, while single depletion of opa or ham leads to either an increase in D$^+$ or Grh$^+$ cells, respectively (*Figure 2*; *Eroglu et al., 2014*). Co-expressing opa RNAi with ham shmiR in an INP-specific manner caused supernumerary Dpn$^+$, Ase$^+$ INPs (*Figure 6—figure supplement 1A*). In addition, the number of D$^+$/Dpn$^+$ and Grh$^+$/Dpn$^+$ INPs were also increased, which is in contrast to single depletion of opa or ham (*Figure 6A–B*, *Figure 2*; *Eroglu et al., 2014*). Thus, opa and ham loss-of-function phenotypes are additive. Importantly, despite inducing over-proliferation of mature INPs (Ase$^+$/Dpn$^+$), depleting both opa and ham in type II NBs could not

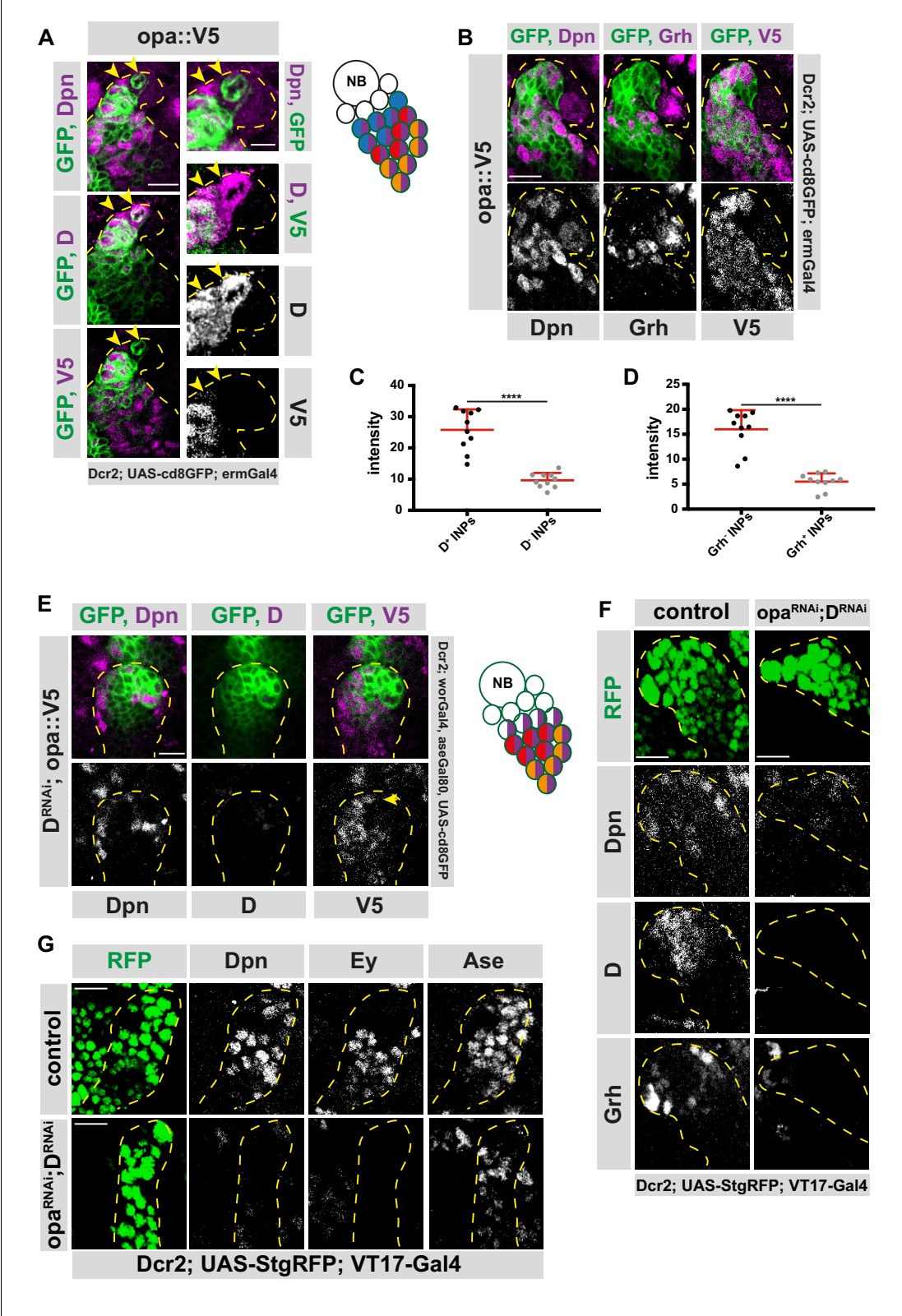

**Figure 4.** Osa initiates D expression before initiating Opa. (**A**) Close-up images of fly brains endogenously expressing V5-tagged opa in INPs, stained for V5, Dpn and D. D⁺, V5⁻ cell is marked with arrows, lineages are outlined with yellow dashed line, scale bar 10 µm and 5 µm, (induced with ermGal4, marked with membrane bound GFP). (**B**) Close-up images of fly brains endogenously expressing V5-tagged opa in INPs, stained for V5, Dpn and Grh, lineages are outlined with yellow dashed line, scale bar 10 µm, (induced with ermGal4, marked with membrane bound GFP). (**C**) Quantifications of opa::

*Figure 4 continued on next page*

*Figure 4 continued*

V5-signal intensity measurements of D$^+$ vs D$^-$ INPs, n = 10, normalized to background intensity. Data represent mean ± SD, ***p<=0.001, Student's t-test. (D) Quantifications of opa::V5-signal intensity measurements of Grh$^+$ vs Grh$^-$ INPs, n = 10, normalized to background intensity. Data represent mean ± SD, ***p<=0.001, Student's t-test. (E) Close-up images of fly brains endogenously expressing V5-tagged opa and RNAi for D in type II lineages, stained for V5, Dpn and D, lineages are outlined with yellow dashed line, scale bar 10 μm, (induced with worGal4, aseGal80, marked with membrane bound GFP). (F–G) Close up images of control versus opa and D double knock-down brains in type II lineages, stained with Dpn, D and Grh (C), or for Dpn, Ey and Ase (C) antibodies, lineages are outlined with yellow dashed lines, scale bar 10 μm, (induced with Dcr2; UAS-StgRFP; VT17-Gal4, marked with nuclear RFP).

DOI: https://doi.org/10.7554/eLife.46566.022

The following source data and figure supplements are available for figure 4:

**Source data 1.** Quantification of intensity measurements of opa::V5 signal in D$^+$ versus D$^-$ INPs in wild-type brains (*Figure 4C*).
DOI: https://doi.org/10.7554/eLife.46566.026

**Source data 2.** Quantification of intensity measurements of opa::V5 signal in Grh$^+$ versus Grh$^-$ INPs in wild-type brains (*Figure 4C*).
DOI: https://doi.org/10.7554/eLife.46566.027

**Figure supplement 1.** Opa is expressed in type II lineages.
DOI: https://doi.org/10.7554/eLife.46566.023

**Figure supplement 2.** Osa initiates the expression of opa in INPs.
DOI: https://doi.org/10.7554/eLife.46566.024

**Figure supplement 3.** Different temporal states have different opa levels.
DOI: https://doi.org/10.7554/eLife.46566.025

**Figure supplement 3—source data 1.** Quantification of intensity measurements of opa::V5 signal in Ey$^+$ versus Ey$^-$ INPs in wild-type brains (*Figure 4—figure supplement 3B*).
DOI: https://doi.org/10.7554/eLife.46566.028

recapitulate the Osa loss-of-function phenotype because imINPs could mature and express Ase, and therefore did not revert into ectopic NBs (*Figure 6—figure supplement 1B*). This suggests that Osa regulates temporal patterning in two levels: initiation by D activation, and progression by opa and ham.

## Discussion

Temporal patterning is a phenomenon where NSCs alter the fate of their progeny chronologically. Understanding how temporal patterning is regulated is crucial to understanding how the cellular complexity of the brain develops. Here, we present a novel, FACS-based approach that enabled us to isolate distinct temporal states of neural progenitors with very high purity from Drosophila larvae. This allowed us to study the transitions between different temporal identity states. We identified odd-paired (opa), a transcription factor that is required for INP temporal patterning. By studying the role of this factor in temporal patterning, we propose a novel model for the regulation of temporal patterning in *Drosophila* neural stem cells.

We establish two different roles of the SWI/SNF complex subunit, Osa, in regulating INP temporal patterning. Initially, Osa initiates temporal patterning by activating the transcription factor D. Subsequently, Osa regulates the progression of temporal patterning by activating opa and ham, which in turn downregulate D and Grh, respectively (*Figure 6C*). The concerted, but complementary action of opa and ham ensures temporal identity progression by promoting the transition between temporal stages. For instance, opa regulates the transition from D to Grh, while ham regulates the transition from Grh to Ey. We propose that opa achieves this by repressing D and activating grh, as indicated by the lack of temporal patterning in D and opa-depleted INPs (*Figure 4C–D*, *Figure 6C*). Loss of opa or ham causes INPs to lose their temporal identity and overproliferate. Moreover, we propose that D and opa activate Grh expression against the presence of ham, which represses Grh expression. As D and opa levels decrease as INPs age and become Grh positive, ham is capable of repressing Grh later on in temporal patterning (*Figure 6C*). This explains how opa and ham act only during specific stages even though they are expressed throughout the entire lineage.

An open question pertains to the fact that the double knock-down of opa and ham, as well as that of D and opa, failed to recapitulate the Osa phenotype. Even though opa and ham RNAi caused massive overproliferation in type II lineages, we could not detect any Dpn$^+$ Ase$^-$ ectopic NB-like cells

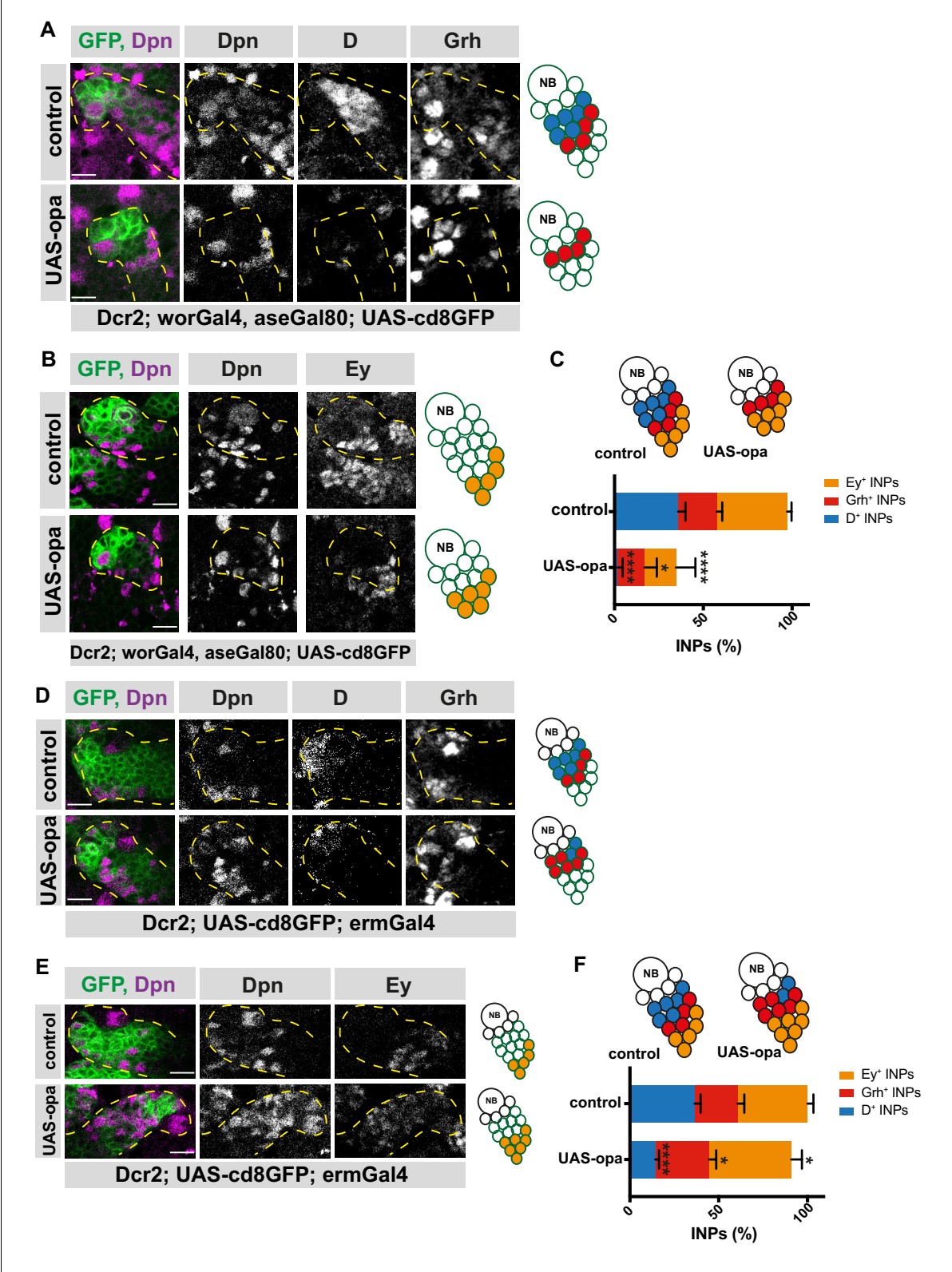

**Figure 5.** Opa overexpression results in the loss of D⁺INPs. (**A**) Close-up images of control and opa overexpressing brains in type II lineages, stained for Dpn, D and Grh antibodies, lineages are outlined with yellow dashed lines, scale bar 10 μm, (induced with worGal4, aseGal80, marked with membrane bound GFP). Overexpression of opa in type II lineages causes the loss of D⁺ INPs. (**B**) Close-up images of control and opa overexpressing brains in type II lineages, stained for Dpn, and Ey antibodies, lineages are outlined with yellow dashed lines, scale bar 10 μm, (induced with worGal4,

*Figure 5 continued on next page*

*Figure 5 continued*

aseGal80, marked with membrane bound GFP). (**C**) Quantification of D$^+$, Grh$^+$ and Ey$^+$ INPs in control and opa overexpressing brains, n = 10, total INP numbers in control were normalized to 100%. Data represent mean ± SD, p<=0.05, ***p<=0.001, Student's t-test (D$^+$ INPs control 12.18 ± 1.33 [n = 10], opa GOF 0.4 ± 0.6 [n = 10], p<0.001; Grh$^+$ INPs control 7.38 ± 1 [n = 10], opa GOF 5.12 ± 2.20 [n = 10], p<0.05; Ey$^+$ INPs control 13.5 ± 0.76 [n = 10], opa GOF 6 ± 3.5 [n = 10], p<0.001). (**D**) Close-up images of control and opa overexpressing brains in INPs, stained for Dpn, and Ey, lineages are outlined with yellow dashed lines, scale bar 10 µm, (induced with ermGal4, marked with membrane bound GFP). (**E**) Close-up images of control and opa overexpressing brains in INPs, stained for Dpn, D and Grh, lineages are outlined with yellow dashed lines, scale bar 10 µm, (induced with ermGal4, marked with membrane bound GFP). (**F**) Quantification of D$^+$, Grh$^+$ and Ey$^+$ INPs in control and opa overexpressing brains, n = 5, total INP numbers in control were normalized to 100%. Data represent mean ± SD, *p<=0.05, ***p<0.001, Student's t-test (D$^+$ INPs control 12.4 ± 1.01 [n = 5], opa GOF 4.83 ± 0.68 [n = 5], p<0.0001; Grh$^+$ INPs control 8.2 ± 1.16 [n = 5], opa GOF 10.33 ± 1.24 [n = 5], p<0.05; Ey$^+$ INPs control 13.4 ± 1.01 [n = 5], opa GOF 15.71 ± 1.9 [n = 5], p<0.05).

DOI: https://doi.org/10.7554/eLife.46566.029

The following source data and figure supplements are available for figure 5:

**Source data 1.** Quantification of number of INPs in three different temporal identities between control versus opa-overexpressed brains with type II-specific driver (*Figure 5C*).
DOI: https://doi.org/10.7554/eLife.46566.032

**Source data 2.** Quantification of number of INPs in three different temporal identities between control versus opa-overexpressed brains with INP-specific driver (*Figure 5F*).
DOI: https://doi.org/10.7554/eLife.46566.033

**Figure supplement 1.** Opa overexpression causes shorter type II lineages.
DOI: https://doi.org/10.7554/eLife.46566.030

**Figure supplement 2.** Opa overexpression causes loss of D$^+$INPs in DM1 lineages.
DOI: https://doi.org/10.7554/eLife.46566.031

**Figure supplement 2—source data 1.** Quantification of number of INPs in three different temporal identities between control versus opa-overexpressed brains with INP-specific driver in DM1 lineages (*Figure 5—figure supplement 2C*).
DOI: https://doi.org/10.7554/eLife.46566.034

(as occurs in Osa mutant clones, *Eroglu et al., 2014*). We propose that this is caused by D expression which is still induced even upon opa/ham double knockdown, but not upon Osa knock-down where D expression fails to be initiated. Thus, the initiation of the first temporal identity state may block the reversion of INPs to a NB-state. In the future, it will be important to understand the exact mechanisms of how opa regulates temporal patterning.

We further demonstrate that Osa initiates D expression earlier than opa expression. Osa is a subunit of SWI/SNF chromatin remodeling complex, and it guides the complex to specific loci throughout the genome, such as the TSS of both D and opa. The differences in timing of D and opa expression may be explained by separate factors involved in their activation. Previous work suggests that the transcription factor earmuff may activate (*Janssens et al., 2014*; *Janssens et al., 2017*). However, it remains unknown which factor activates opa expression. One possibility is that the cell cycle activates opa, since its expression begins in mINPs, a dividing cell unlike imINPs, which are in cell cycle arrest.

We propose that balanced expression levels of D and opa regulates the timing of transitions between temporal identity states. Indeed, Osa initiates D and opa, the repressor of D, at slightly different times, which could allow a time window for D to be expressed, perform its function, then become repressed again by opa. Deregulating this pattern, for example by overexpressing opa in the earliest INP stage, results in a false start of temporal patterning and premature differentiation. This elegant set of genetic interactions resembles that of an incoherent feedforward loop (FFL) (*Kim et al., 2008*; *Mangan and Alon, 2003*). In such a network, pathways have opposing roles. For instance, Osa promotes both the expression and repression of D. Similar examples can be observed in other organisms, such as in the galactose network of *E. coli*, where the transcriptional activator CRP activates galS and galE, while galS also represses galE (*Shen-Orr et al., 2002*). In Drosophila SOP determination, miR-7, together with Atonal also forms an incoherent FFL (*Li et al., 2009*). Furthermore, mammals apply a similar mechanism in the c-Myc/E2F1 regulatory system (*O'Donnell et al., 2005*).

The vertebrate homologues of opa consist of the Zinc-finger protein of the cerebellum (ZIC) family, which are suggested to regulate the transcriptional activity of target genes, and to have a role in CNS development (*Elms et al., 2004*; *Elms et al., 2003*; *Gaston-Massuet et al., 2005*; *Inoue et al.,*

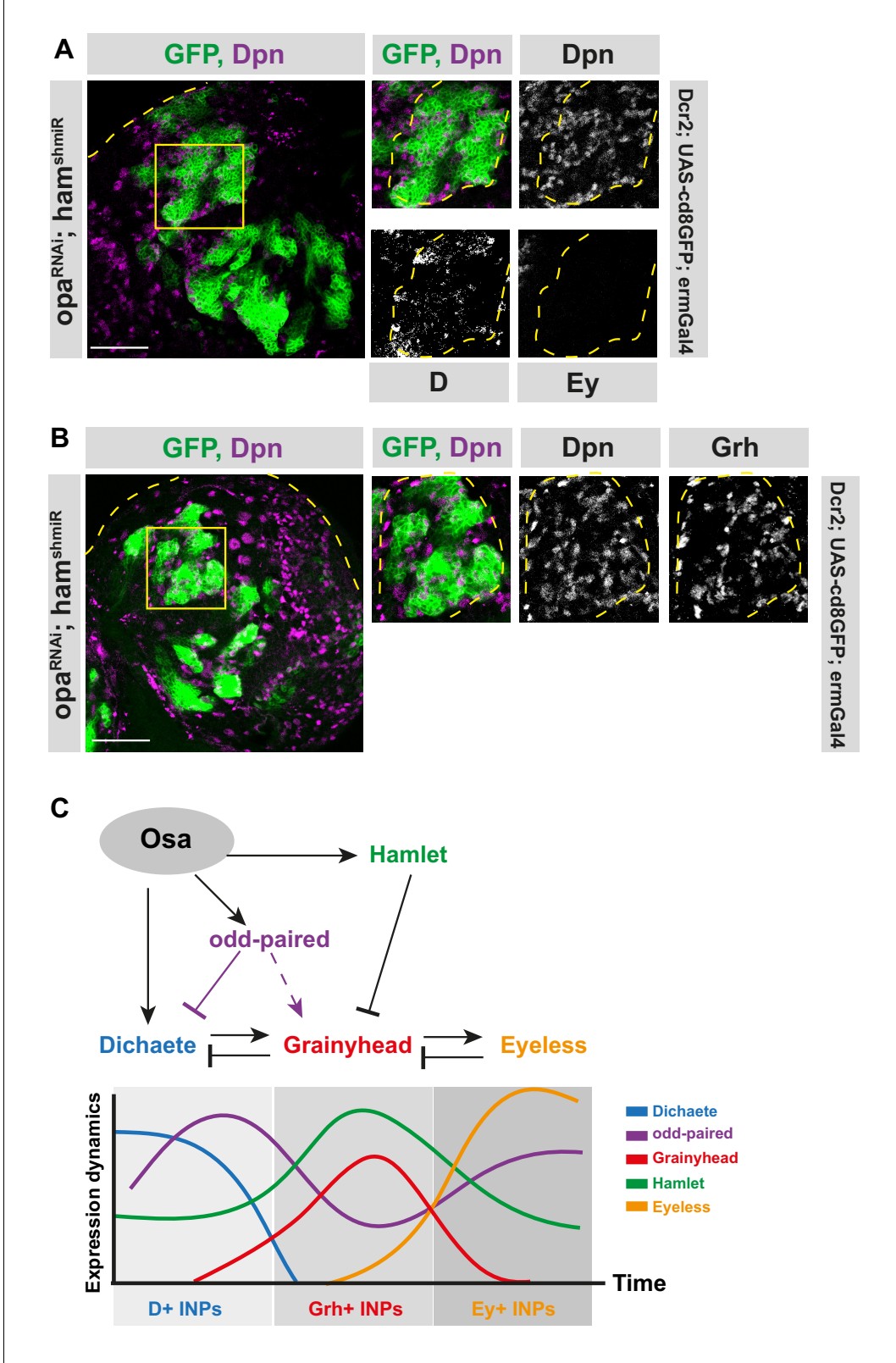

**Figure 6.** Opa and hamlet are required for INP temporal patterning and correct lineage progression. (A–B) Overview images of brain lobes expressing RNAi against opa and ham in INPs and their close-up images (marked with yellow squares), stained for Dpn, D and Ey (A), or Dpn and Grh (B) antibodies, lineages and lobes are outlined with yellow dashed lines, scale bar 50 µm for brain lobes, 10 µm for zoomed images, (induced with

*Figure 6 continued*

ermGal4, marked with membrane bound GFP). (**C**) Model depicting the genetic interactions between temporal switch genes (opa and hamlet), and temporal identity genes (D, Grh, and Ey).

DOI: https://doi.org/10.7554/eLife.46566.035

The following figure supplement is available for figure 6:

**Figure supplement 1.** Opa and hamlet cannot recapitulate Osa knock-down phenotype.

DOI: https://doi.org/10.7554/eLife.46566.036

*2004*; *Inoue et al., 2007*). In mice, during embryonic cortical development, ZIC family proteins regulate the proliferation of meningeal cells, which are required for normal cortical development (*Inoue et al., 2008*). In addition, another member of the ZIC family, Zic1, is a Brn2 target, which itself controls the transition from early-to-mid neurogenesis in the mouse cortex (*Urban et al., 2015*). Along with these lines, it has been shown that ZIC family is important in brain development in zebrafish (*Maurus and Harris, 2009*; *Sanek and Grinblat, 2008*). Furthermore, the role of ZIC has been implicated in variety of brain malformations and/or diseases (*Aruga et al., 2010*; *Blank et al., 2011*; *Hatayama et al., 2011*). These data provide mere glimpses into the roles of ZIC family proteins in neuronal fate decisions in mammals, and our study offers an important entry point to start understanding these remarkable proteins.

Our findings provide a novel regulatory network model controlling temporal patterning, which may occur in all metazoans, including humans. In contrast to existing cascade models, we instead show that temporal patterning is a highly coordinated ensemble that allows regulation on additional levels than was previously appreciated to ensure a perfectly balanced generation of different neuron/glial cell types. Together, our results demonstrate that *Drosophila* is a powerful system to dissect the genetic mechanisms underlying the temporal patterning of neural stem cells and how the disruption of such mechanisms impacts brain development and behavior.

# Materials and methods

## Key resources table

| Reagent type (species) or resource | Designation | Source or reference | Identifiers | Additional information |
|---|---|---|---|---|
| Gene (*Drosophila melanogaster*) | osa | NA | FBgn0261885 | |
| Gene (*D. melanogaster*) | Dichaete | NA | FBgn0000411 | |
| Gene (*D. melanogaster*) | Grainyhead | NA | FBgn0259211 | |
| Gene (*D. melanogaster*) | Eyeless | NA | FBgn0259211 | |
| Gene (*D. melanogaster*) | Hamlet | NA | FBgn0045852 | |
| Gene (*D. melanogaster*) | Odd-paired | NA | FBgn0003002 | |
| Genetic reagent (*D. melanogaster*) | UAS-CD8::GFP; *erm*GAL4 | PMID:18621688 and 20152183 | | |
| Genetic reagent (*D. melanogaster*) | UAS-CD8::td Tomato; *erm*GAL4 | PMID:18621688 and 20152183 | | |
| Genetic reagent (*D. melanogaster*) | UAS-*dcr2*; *wor*-GAL4, *ase*GAL80; UAS-CD8::GFP | PMID:21549331 | | |
| Genetic reagent (*D. melanogaster*) | *VT17*-GAL4 | Vienna Drosophila RNAi Center | 212057, discarded | |
| Genetic reagent (*D. melanogaster*) | UAS-*stinger*::RFP | PMID:11056799 | | |

*Continued on next page*

Continued

| Reagent type (species) or resource | Designation | Source or reference | Identifiers | Additional information |
|---|---|---|---|---|
| Genetic reagent (*D. melanogaster*) | UAS-*opa*<sup>RNAi</sup> | Vienna Drosophila RNAi Center | 101531 | |
| Genetic reagent (*D. melanogaster*) | UAS-D<sup>RNAi</sup> | Vienna Drosophila RNAi Center | 49549 and 107194 | |
| Genetic reagent (*D. melanogaster*) | UAS-mcherry<sup>shmiR</sup> | Bloomington Drosophila Stock Center | 35785 | |
| Genetic reagent (*D. melanogaster*) | UAS-osa<sup>RNAi</sup> | Vienna Drosophila RNAi Center | 7810 | |
| Genetic reagent (*D. melanogaster*) | UAS-ham<sup>shmiR</sup> | Bloomington Drosophila Stock Center | 32470 | |
| Genetic reagent (*D. melanogaster*) | UAS-osa<sup>shmiR</sup> | PMID:2460726 | | |
| Genetic reagent (*D. melanogaster*) | UAS-p35 | PMID:7925015 | | |
| Genetic reagent (*D. melanogaster*) | UAS-opa | PMID:17329368 | | |
| Genetic reagent (*D. melanogaster*) | D::GFP | this paper | | endogenously GFP-tagged D in C-terminus |
| Genetic reagent (*D. melanogaster*) | Grh-GFP | Bloomington Drosophila Stock Center | 42272 | |
| Genetic reagent (*D. melanogaster*) | Ey-GFP | Bloomington Drosophila Stock Center | 42271 | |
| Genetic reagent (*D. melanogaster*) | opa::V5 | this paper | | endogenously V5-tagged opa in C-terminus |
| Genetic reagent (*D. melanogaster*) | FRT82B, *opa*<sup>7</sup> | PMID:17329368 | | |
| Genetic reagent (*D. melanogaster*) | *elav*Gal4 (C155) | PMID:10197526 | | |
| Genetic reagent (*D. melanogaster*) | actCas9 | Bloomington Drosophila Stock Center | 54590 | |
| Genetic reagent (*D. melanogaster*) | hsCre | Bloomington Drosophila Stock Center | 851 | |
| Antibody | anti-Deadpan (guinea pig, polyclonal) | PMID:2460726 | | (1:1000) |
| Antibody | anti-Asense (rat, polyclonal) | PMID:2460726 | | (1:500) |
| Antibody | anti-Miranda (guinea pig, polyclonal) | PMID:2460726 | | (1:500) |
| Antibody | anti-Grainyhead (rat, polyclonal) | PMID:19945380 | | (1:1000) |
| Antibody | anti-Dichaete (rabbit, polyclonal) | gift from Steve Russell | | (1:1000) |
| Antibody | anti-Eyeless (mouse, monoclonal) | Developmental Studies Hybridoma Bank | anti-eyeless | (1:50), RRID:AB_2253542 |
| Antibody | anti-Toy (guinea pig, polyclonal) | gift from Uwe Walldorf | | (1:500) |

| Reagent type (species) or resource | Designation | Source or reference | Identifiers | Additional information |
|---|---|---|---|---|
| Antibody | anti-Bsh (guinea pig, polyclonal) | gift from Makoto Sato, PMID:21303851 | | (1:500), RRID:AB_2567934 |
| Antibody | anti-V5 (mouse, monoclonal) | Sigma Aldrich | V8012 | (1:500 IF, 1:1000 WB), RRID:AB_261888 |
| Antibody | anti-Bruchpilot nc82 (mouse, monoclonal) | Developmental Studies Hybridoma Bank | nc82 | (1:10), RRID:AB_2314866 |
| Antibody | anti-V5 IgG2a (mouse, monoclonal) | Thermo Fisher Scientific | R960-25 | (1:500), RRID:AB_2556564 |
| Antibody | anti-V5 (rabbit, polyclonal) | Abcam | ab9116 | (1:500), RRID:AB_307024 |
| Antibody | anti-Prospero (mouse, monoclonal) | Developmental Studies Hybridoma Bank | MR1A | (1:20), RRID:AB_528440 |
| Antibody | anti-pH3(Ser10) (mouse, monoclonal) | Cell Signaling Technologies | 9706S | (1:1000), RRID:AB_331748 |
| Antibody | anti-aPKC (rabbit, polyclonal) | Santa Cruz Biotechnologies | sc-216 | (1:500), RRID:AB_2300359 |
| Antibody | anti-alpha tubulin (mouse, monoclonal) | Sigma Aldrich | T6199 | (1:10000), RRID:AB_477583 |
| Antibody | Alexa 405, 568, 647 | Invitrogen | Alexa Fluor dyes | (1:500) |
| Antibody | IRDye 700, 800 | LI-COR | IRDye | (1:15000) |
| Software, algorithm | Prism 7 | GraphPad Software | | |
| Software, algorithm | BWA | PMID:19451168 | | RRID:SCR_010910 |
| Software, algorithm | TopHat | PMID:19289445 | | RRID:SCR_013035 |
| Software, algorithm | HTSeq | PMID:25260700 | | RRID:SCR_005514 |
| Software, algorithm | DESeq2 (v1.12.4) | PMID:25516281 | | RRID:SCR_016533 |
| Software, algorithm | bedtools (v2.26.0) | PMID:20110278 | | RRID:SCR_006646 |
| Commercial assay | TRIzol LS | Ambion | 10296010 | |
| Commercial assay | Agencourt AMPure XP beads | Beckman Coulter | A63880 | |
| Commercial assay | Nextera DNA Library Prep Kit | Illumina | FC-121–1031 | |
| Recombinant DNA reagent | pU6-Bbsl-chiRNA | PMID:23709638 | | |
| Other | Rinaldini solution | PMID:22884370 | | |

## Fly strains, RNAi, and clonal analysis

The following *Drosophila* stocks were used: UAS-opa[RNAi] (VDRC, TID: 101531), UAS-mcherry[shmiR] (BL35785), UAS-D[RNAi] (VDRC, TID: 49549, 107194), UAS-osa[RNAi] (VDRC, TID: 7810), UAS-ham[shmiR] (BL32470), UAS-osa[shmiR] (*Eroglu et al., 2014*), UAS-p35, UAS-opa (*Lee et al., 2007*), PBac{grh-GFP. FPTB}VK00033 (BL42272), PBac{EyGFP.FPTB}VK00033 (BL42271) (*Spokony and White, 2012*), D:: GFP (generated in this study), opa::V5 (generated in this study). GAL4 driver lines used: UAS-cd8:: tdTomato; *erm*Gal4, UAS-cd8::GFP; *erm*Gal4 (*Pfeiffer et al., 2008*; *Weng et al., 2010*), UAS-*dcr2*; *wor*Gal4, *ase*Gal80; UAS-cd8::GFP (*Neumüller et al., 2011*), UAS-*dcr2*; UAS-cd8::GFP; VT17-Gal4

(VDRC, TID: 212057, discarded). Mutant fly strains used for clonal analysis were FRT82B, *opa*[7] (*Lee et al., 2007*). Clones were generated by Flippase (FLP)/FLP recombination target (FRT)-mediated mitotic recombination, using the *elav*Gal4 (C155) (*Lee and Luo, 1999*). Larvae were heat shocked for 90 min at 37°C and dissected as third-instar wandering larvae (120 hr). RNAi crosses were set up and reared at 29°C, and five days later, third-instar wandering larvae were dissected. *w*[118] was used as control for comparison with RNAi lines, whereas UAS-mcherry[shmiR] was used as control for comparison with shmiR lines, and experiments involving UAS-transgenes.

## Generation of opa::V5 and D::GFP flies

For both genes, the guides were cloned as overlapping oligos into linearized pU6-BbsI-chiRNA (Addgene 45946, *Gratz et al., 2013*) and injected at 100 ng/µl into actCas9 flies (BL 54590, *Port et al., 2014*). Donors (either oligos or plasmid) were co-injected at 250 ng/µl. For opa, donors were Ultramer Oligos from IDT with around 60nt homology arms on either side. For D, homology arms were 800 bp and 900 bp long. Donor plasmid contained GFP, V5, 3xFlag, and dsRed. They were screened for dsRed eyes and then, the selection cassette was removed with hsCre (BL 851). *opa* gRNA GATGCATCCCGGCGCAGCGA *opa* donor GAACCCGCTGAACCATTTCGGACACCA TCACCACCACCACCACCTGATGCATCCCGGCGCgGCaACcGCGTATggtaagcctatacc- taaccctcttcttggTCTAGAtagcacgTGAGAGTGGGAGAACTGG TGGCCCGAGGAGGCGCCACCGCCGGCCGCCCAACCGA

   *D* gRNA GTGCTCTATTAGAGTGGAGT

## Negative geotaxis assay

Negative geotaxis assay was used as described before (*Ali et al., 2011*), where the percentage of flies passing the 8.5 cm mark in 10 s was assessed. For each genotype and gender, 10 two-day old adult flies in 10 biological replicates were measured and for each replicate, 10 measurements were performed with 1 min rest period in between.

## Immunohistochemistry and antibodies

Larval or adult brains were dissected in 1X PBS, and then fixed for 20 min at room temperature (RT) in 5% paraformaldehyde in PBS and washed once with 0.1% TritonX in PBS (PBST). The brains were incubated for 1 hr at RT with blocking solution (5% normal goat serum or 1% BSA in PBST). Blocking was followed by overnight incubation at 4°C with primary antibodies in blocking solution. Then, the brains were washed three times with PBST, and incubated for 1 hr at RT with secondary antibodies (1:500, goat Alexa Fluor, Invitrogen) in blocking solution. After secondary antibody, brains were washed three times with PBST, and mounted in Vectashield Antifade Mounting Medium (Vector Labs).

   Antibodies used in this study were: guinea pig anti-Deadpan (1:1000, *Eroglu et al., 2014*), rat anti-Asense (1:500, *Eroglu et al., 2014*), guinea pig anti-Miranda (1:500, *Eroglu et al., 2014*), rat anti-Grh (1:1,000; *Baumgardt et al., 2009*); rabbit anti-D (1:1,000; gift from Steve Russell); mouse anti-Ey (1:10; DSHB); guinea pig anti-Toy (gift from Uwe Walldorf), guinea pig anti-Bsh (gift from Makoto Sato), mouse anti-Bruchpilot nc82 (1:10, DSHB), mouse anti-V5 (1:500, Sigma Aldrich, V8012), mouse antiV5 IgG2a (Thermo Fisher Scientific, R960-25, used in *Figure 4—figure supplement 1D*), rabbit anti-V5 (Abcam, ab9116, used in *Figure 4—figure supplement 3A*), mouse anti-Pros (1:100, Developmental Studies Hybridoma Bank), mouse anti-pH3(Ser10) (1:500, Cell Signaling Technologies, 9701S), rabbit anti-aPKC (1:500, Santa Cruz Biotechnology, sc-216). Throughout the paper, for every quantification, dorsomedial 2 and 3 type II NB lineages (DM2 and 3) were considered, if not stated otherwise.

## In vitro immunofluorescence

FACS-sorted cells from ~300 larval brains (UAS-cd8::tdTomato, *erm*Gal4) or their unsorted control matches were plated on cover glass (Labtek II Chambered Coverglass, 8-well, 155409, Thermo Fisher Scientific) into Schneider's medium (*Homem et al., 2013*). The dishes were placed onto ice and cells were incubated for 1 hr to settle down. Cells were then fixed with 5% PFA in PBS at RT and washed three times with 0.1% PBST. After washes, cells were incubated for 1 hr at RT with blocking solution (5% normal goat serum in 0.1% PBST). The cells were then incubated overnight at

4°C with primary antibodies in blocking solution, which was followed by three washes with 0.1% PBST, and secondary antibody (1:500, goat Alexa Fluor, Invitrogen) incubation for 1 hr at RT. Cells were again washed three times with 0.1% PBST, and then mounted in in Vectashield Antifade Mounting Medium with Dapi (Vector Labs).

## Microscopy
Confocal images were acquired with Zeiss LSM 780 confocal microscopes.

## Western blot
Embryos were collected and dechorionated, then boiled in 2x Laemmli buffer and loaded on 4–12% gradient Bis-Tris gels (NuPAGE, Invitrogen). After SDS-PAGE according to Invitrogen's protocol, proteins were transferred to a Nitrocellulose membrane (0.22 µm, Odyssey LI-COR) for 2 hr at 100V, blocked with 5% milk powder in blocking solution (PBS with 0.2% Tween) for 1 hr, overnight incubation with primary antibody in blocking solution at 4°C, 3x washed with washing solution (PBS with 0.1% Tween) and followed by 1 hr incubation with secondary antibody (1:15000, goat IRDye, LI-COR)in blocking solution. After three washes with washing solution, the membranes were air-dried, and fluorescent signal were detected with Odyssey CLx imaging system (Odyssey CLx LI-COR). Antibodies used were: mouse anti-V5 (1:1000, Sigma Aldrich, V8012), anti-alpha tubulin (1:10000, Sigma Aldrich, T6199).

## Intensity measurements
For intensity measurements of opa-V5 signal, cells expressing Dpn and temporal identity markers (D, Grh or Ey) were circled with selection tools. Raw integrity density (sum of gray values of all selected pixels) was measured using FIJI. In each image, five temporal identity positive INP and five temporal identity negative INP were measured for raw integrity density along with three background circles with no opa-V5 signal, (eg. $D^+$ vs $D^-$ INPs). Then, corrected total cell fluorescent (CTCF) were calculated with 'Integrated density – (Area of selected cells X Mean fluorescence of background readings)' (*McCloy et al., 2014*). Then, the mean of temporal identity positive versus negative cells were calculated and the values were normalized to means of background for each brain.

## Statistics
Statistical analyses were performed with GraphPad Prism 7. Unpaired two-tailed Student's *t*-test was used to assess statistical significance between two genotypes. Experiments were not randomized, and investigator was not blinded. Sample sizes for experiments were estimated on previous experience with similar setup which showed significance, thus, no statistical method was used to determine sample size.

## Cell dissociation and FACS
Cell dissociation and FACS were performed as previously described with minor changes (*Berger et al., 2012*; *Harzer et al., 2013*). UAS-cd8::tdTomato; *erm*Gal4 driver line was used to induce expression of membrane bound tdTomato in INPs. In addition to the driver lines, temporal identity factors were tagged with GFP. Flies expressing both fluorophores were dissected at L3 stage, and then dissociated into single cell suspension. Decreasing levels of tdTomato were observed in differentiated cells due to lack of driver line expression. Thus, biggest cells with highest tdTomato expression and highest GFP expression were sorted.

For RNA isolation, cells were sorted directly in TRIzol LS (10296010, Invitrogen), while for cell staining, they were sorted on coated glass-bottomed dishes and stained as previously described (*Berger et al., 2012*).

## RNA isolation, cDNA synthesis and qPCR
RNA was isolated using TRIzol LS reagent (10296010, Invitrogen) from FACS sorted cells. Then RNA samples were used as template for first-strand cDNA synthesis with random hexamer primers (SuperScriptIII, Invitrogen). qPCR was done using Bio-Rad IQ SYBR Greeen Supermix on a Bio-Rad CFX96 cycler. Expression of each gene was normalized to Act5c, and relative levels were calculated using the $2^{-\Delta\Delta CT}$ method (*Livak and Schmittgen, 2001*). Primer used were:

*act5c* AGTGGTGGAAGTTTGGAGTG, GATAATGATGATGGTGTGCAGG
*D* ATGGGTCAACAGAAGTTGGGAG, GTATGGCGGTAGTTGATGGAATG
*grh* TCCCCTGCTTATGCTATGACCT, TACGGCTAGAGTTCGTGCAGA
*ey* TCGTCCGCTAACACCATGA, TGCTCAAATCGCCAGTCTGT
*ham* ATAGATCCTTTGGCCAGCAGAC, AGTACTCCTCCCTTTCGGCAAT
*opa* CTGAACCATTTCGGACACCATC, CCAGTTCTCCCACTCTCAATAC

## RNA sequencing – DigiTAG

For each experiment 6000–7000 FACS-sorted D$^+$, Grh$^+$ or Ey$^+$ INPs were isolated by TRIzol purification. Three replicates from each temporal state were analyzed. RNA samples were reverse transcribe into first-strand cDNA using SuperScriptIII Reverse Transcriptase (Invitrogen) with oligo-(dT)2- primers. Then the second-strand cDNA were generated. It was followed by library preparation with Nextera DNA Library Preparation Kit (Illumina) as previously described (*Landskron et al., 2018*; *Wissel et al., 2018*). Libraries were purified with Agencourt AMPure XP beads. Purified libraries were then subjected to 50 base pair Illumina single-end sequencing on a Hiseq2000 platform.

## Transcriptome data analysis

### Alignment

Unstranded reads were screened for ribosomal RNA by aligning with BWA (v0.7.12; *Li and Durbin, 2009*) against known rRNA sequences (RefSeq). The rRNA subtracted reads were aligned with TopHat (v2.1.1; *Kim et al., 2013*) against the Drosophila genome (FlyBase r6.12). Introns between 20 and 150,000 bp are allowed, which is based on FlyBase statistics. Microexon-search was enabled. Additionally, a gene model was provided as GTF (FlyBase r6.12).

### Deduplication

Reads arising from duplication events are marked as such in the alignment (SAM/BAM files) as follows. The different tags are counted at each genomic position. Thereafter, the diversity of tags at each position is examined. First, tags are sorted descending by their count. If several tags have the same occurrence, they are further sorted alphanumerically. Reads sharing the same tag are sorted by the mean PHRED quality. Again, if several reads have the same quality, they are further sorted alphanumerically. Now the tags are cycled through by their counts. Within one tag, the read with the highest mean PHRED quality is the unique cor- rect read and all subsequent reads with the same tag are marked as duplicates. Furthermore, all reads that have tags with one mis- match difference compared the pool of valid read tags are also marked as duplicates.

### Summarization

Small nuclear RNA, rRNA, tRNA, small nucleolar RNA, and pseudogenes are masked from the GTF (FlyBase r6.12) with subtractBed from bedtools (v2.26.0; *Quinlan and Hall, 2010*). The aligned reads were counted with HTSeq (v0.6.1; intersec- tion-nonempty), and genes were subjected to differential expres- sion analysis with DESeq2 (v1.12.4; *Love et al., 2014*).

## Hierarchical clustering analysis

Genes are filtered by the indicated log2fc and an adjusted P value < 0.05 in at least one pairwise comparison. In addition, a minimal expression of 10 RPM in at least one condition was required. The tree cut into four clusters (different cluster numbers were tested; *Kolde and Package, 2015*, 202AD). GO analysis was performed with FlyMine (*Lyne et al., 2007*), Holm-Bonferroni correction with max p-value 0.05 was used. Biological process and molecular function were the ontologies.

## Accession numbers

The Gene Expression Omnibus accession number for the RNA-sequencing data reported in this paper is GSE127516.

## GO-term analysis

Gene Ontology (GO) enrichment analysis were performed on www.flymine.org/with Holm-Bonferroni correction with max p-value 0.05. Biological process and molecular function were the ontologies.

## Acknowledgements

We thank all Knoblich lab members for support and discussions, Francois Bonnay, Tom Kruitwagen and Joshua A Bagley for comments on the manuscript, Peter Duchek, Joseph Gokcezade, Elke Kleiner, the IMP/IMBA Biooptics Facility and the Next Generation Sequencing Unit of the Vienna Biocenter Core Facilities (VBCF) for assistance and Makoto Sato, Uwe Walldorf, Stefan Thor and Steve Russel for sharing reagents, and the Harvard TRiP collection, the Bloomington Drosophila stock center and the Vienna Drosophila Resource Center (VDRC) for reagents.

## Additional information

### Funding

| Funder | Grant reference number | Author |
| --- | --- | --- |
| Austrian Academy of Sciences | | Jürgen A Knoblich |
| Austrian Science Fund | Z_153_B09 | Jürgen A Knoblich |
| European Commission | | Jürgen A Knoblich |

The funders had no role in study design, data collection and interpretation, or the decision to submit the work for publication.

### Author contributions

Merve Deniz Abdusselamoglu, Conceptualization, Formal analysis, Investigation, Visualization, Writing—original draft, Project administration; Elif Eroglu, Conceptualization, Data curation, Formal analysis; Thomas R Burkard, Formal analysis, Conducted all bioinformatic analyses; Jürgen A Knoblich, Conceptualization, Supervision, Funding acquisition, Writing—original draft, Project administration, Writing—review and editing

### Author ORCIDs

Merve Deniz Abdusselamoglu (iD) https://orcid.org/0000-0001-8467-3947
Jürgen A Knoblich (iD) https://orcid.org/0000-0002-6751-3404

### Decision letter and Author response
Decision letter https://doi.org/10.7554/eLife.46566.041
Author response https://doi.org/10.7554/eLife.46566.042

## Additional files

### Supplementary files
• Supplementary file 1. List of genes that are dynamically changing between INP populations with higher expression in $D^+$ and $Ey^+$ INPs.
DOI: https://doi.org/10.7554/eLife.46566.037

• Supplementary file 2. List of genes that are dynamically changing between INP populations with highest expression in $Grh^+$ INPs.
DOI: https://doi.org/10.7554/eLife.46566.038

• Transparent reporting form
DOI: https://doi.org/10.7554/eLife.46566.039

### Data availability
Sequencing data have been deposited in GEO under accession code GSE127516. All data generated or analyzed during this study are included in the manuscript and supporting files. Source data files have been provided for Fig1, Fig1supp1, Fig2, Fig2supp1, Fig2supp2, Fig3, Fig4, Fig4supp3, Fig5 and Fig5supp2.

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
