## [Decision Letter]

Thank you for submitting your article "Transcription factor Odd-paired regulates temporal identity in neural progenitors via an incoherent feed-forward loop" for consideration by *eLife*. Your article has been reviewed by three reviewers and the evaluation has been overseen by K VijayRaghavan as the Senior and Reviewing Editor. The following individuals involved in review of your submission have agreed to reveal their identity: Hongyan Wang (Reviewer #1); Sonia Sen (Reviewer #2).

The reviewers have discussed the reviews with one another and the Reviewing Editor has drafted this decision to help you prepare a revised submission. This a very nice study with valuable conclusions. Congratulations.

Summary:

In the *Drosophila* brain, Type II neuroblasts (NBs) generate transit-amplifying lineages (INPs) that divide like Type I NBs to expand the neural population. Like the NBs, INPs too experience temporal patterning allowing them to generate different progeny over time. The temporal cascade in INPs consists of D>Grh>Ey and earlier work from this lab had uncovered the mechanism by which the cascade is initiated in INPs (via Osa activating D), and the mechanism by which the Grh>Ey temporal switch occurred (via Ham repressing Grh). However, much remains unknown in INP temporal patterning.

In this work, Abdusselamoglu et al., take a transcriptomic approach to this problem. They use FACS sorting to isolate the three temporally distinct INP populations followed by RNA-seq analysis to identify genes that are differentially expressed at these stages. To identify factors that ensure proper temporal switching in the INPs, the authors focus on Opa, which displays an RNA expression profile complementary to Ham. Using genetic manipulations and immunohistochemistry, they show that Opa is activated by Osa, and is responsible for repressing D (the first temporal factor in the INP). They also show neural fate specification effects that are consistent with this temporal progression.

Overall the data are of high quality, well documented and clearly presented. To my knowledge, this is the first time INP temporal patterning has been revisited since their first descriptions in 2013 and 2014, and so this manuscript advances our understanding in this field. This is particularly true due to the nature of the approach taken and the tools generated in the process.

Before we highlight the major concerns which can be speedily addressed, here are some specific points, appreciating the study, that a reviewer raised, which could be of value to the authors.

Some general comments which may be useful:

The differentially regulated genes in the temporal windows belong to different GO terms, which are difficult to make sense of, except for the presence of glial markers in the later Ey temporal window. This is of reduced importance.

However, the authors then focus on genes that regulate the transitions from one window to the next. The find that opa is a good candidate as a 'target' of Osa, a component of the SWI/SNF complex. Opa has an expression pattern that is anti-correlated with Hamlet. Interestingly, opa mutants appears to stop the progression of the temporal clock at D, and later windows do not open. The authors conclude that opa inhibits D and acts to promote the transition from D to Grh and correspondingly, leads to the accumulation of Bsh early neurons and the loss of late Eya neurons. But then how is opa controlled to inhibit D if both are osa targets?

The most interesting part of the paper deals with the answer to this question and presents a model that osa regulates both opa and Dichaete, but with very different kinetics: they propose that opa represses D, and thus allow the progression of the temporal cascade and the activation of Grh. This is where the authors introduce the notion of an incoherent feed-forward loop in which osa activates both opa and D, but then opa represses D, but only late as it is turned on later than D. This is an excellent motif to allow temporal progression (or circling).

The paper is very solid and makes robust conclusions, and the model that the timing of expression controls the efficiency of their incoherent FFL is well supported and consistent with the data.

Therefore, the work adds one important detail to the concept that temporal windows progress through time. They show that a circuit motif allows the efficient progression of the cascade. Even if opa is not a temporal factor, it plays an important role in the transitions. We believe that this is an important result.

A concern though is the role of osa, which controls everything while it is a chromatin complex that is recruited by transcription factors: what these TFs are, remains a mystery although their identification would be a major breakthrough.

Essential revisions:

1) When expressing D-GFP, Ey-GFP and Grh-GFP for FACS sorting, are they expressed under their endogenous level to avoid overexpression effect? The D-GFP was described in the methods and appeared to be based on CRISPR/Cas9- mediated gene editing. If this is true, please indicate it clearly. There is no description or citation for the generation of Ey-GFP and Grh-GFP.

2) In FACS-purified GrH^+^ INPs (Figure 2F), the expression Ey is also high. What is the reason for this contamination? Does it mean that many INPs have co-expression of Grh and Ey at the same time?

3) The reason for choosing to focus on Opa is unclear. The expression of Ham in D+, GrH^+^ and Ey^+^ INPs are similar, only fluctuating slightly. Therefore, the expression pattern of Ham doesn't seem to be a good example of dynamic expression in INPs. Can the authors clearly indicate how many genes have dynamic expression patterns in INPs and what criteria, i.e. based on fold changes, is applied to rank and select Opa? Currently, it appeared to be hand-picked.

4) Is asymmetric division of INPs impaired and resulting in the change of INPs numbers upon loss-of-opa or opa overexpression?

5) We appreciate the excellent quality of the FACS sorting of the three temporally distinct INP populations and are very impressed with how cleanly the authors were able to isolate the 3 populations, and how well they have documented it! This makes the RNA-seq data particularly invaluable to the field.

6) With respect to Opa and its place in the temporal transitions, we are largely in agreement with the authors and their interpretations. They propose a model where Opa represses D and suggest that Opa also activates Grh. However, in their subsequent interpretations they seem to consider Opa's repression of D as its main mechanism of action in temporal progression. Would not Opa's activation of Grh, leading indirectly to repression of D, tie in all the data from this and the Byaraktar and Eroglu papers better? It would explain why D^+^, Opa^+^ double positive INPs are seen at all, why D-/- INPs can still progress through the temporal cascade, and why overexpressing Opa resulted in loss of D^+^ INPs and increase in GrH^+^, Ey^+^ INPs (incomplete expression of Grh in UAS-Opa shown in Figure 5 might be due to presence of Ham). Furthermore, it would not call upon differential levels of expression of Opa (which the authors have not shown at the protein level), as timing of expression/inhibition of these genes as the INP divides might account for the temporal transitions.

7) If the authors agree with this, have they looked at the DM1 lineage, which is known to not express Grh? Is Opa expressed in this lineage? If not, does misexpression of Opa there result in Grh activation and D repression? The authors have analysed this in DM2 and DM3 – the same brains could be analysed for DM1.

8) The authors must have verified the Opa:V5 tool before using it. Could they please describe this?

9) The authors show that D comes up in INPs before Opa does. However, this is not very clear from the images. Could the authors maybe show magnified insets of these types INPs with the two reporters co-localised? Am I right in understanding that apart from this one difference, Opa is co-localised with every other temporal factor in the INPs? Related to this, the authors find that Opa RNA levels are high early and late in the lineage, and dip in the middle. Do the protein levels reflect these dynamics? Their discussion seems to suggest it does.

---

## [Author Response]

Essential revisions:1) When expressing D-GFP, Ey-GFP and Grh-GFP for FACS sorting, are they expressed under their endogenous level to avoid overexpression effect? The D-GFP was described in the methods and appeared to be based on CRISPR/Cas9- mediated gene editing. If this is true, please indicate it clearly. There is no description or citation for the generation of Ey-GFP and Grh-GFP.

We apologize for this oversight and have now 1) included the reference for Grh and Ey-GFP flies (commercially available stocks through Bloomington *Drosophila* Stock Center) and 2) clarified our description of the D-GFP strain (subsection “Transcriptome analysis of distinct INP temporal states”. Grh-GFP and Ey-GFP strains have been generated through the insertion of C-terminally-tagged BAC-clones (subsection “Transcriptome analysis of distinct INP temporal states”). To exclude potential influence of Grh or Ey extra copies on temporal patterning, we quantified the numbers of each three different INP subpopulations and found no significant difference compared to wild-type. These new data are now included in Figure1—figure supplement1A.

2) In FACS-purified GrH^+^ INPs (Figure 2F), the expression Ey is also high. What is the reason for this contamination? Does it mean that many INPs have co-expression of Grh and Ey at the same time?

We agree with the reviewer that the Ey mRNA levels in GrH^+^ INPs are high at the mRNA level. However, our staining showed that the amount of Ey protein as monitored by IF in GrH^+^ INPs was very low (Figure 1D and Figure 1F). We are confident that both our transcriptome data and IF are correct and that Ey mRNA level in this case does not reflect its protein level. This could further suggest a post-transcriptional regulation of Ey during Grh-to-Ey transition. We have now included this important point in subsection “Transcriptome analysis of distinct INP temporal states”.

3) The reason for choosing to focus on Opa is unclear. The expression of Ham in D^+^, GrH^+^ and Ey^+^ INPs are similar, only fluctuating slightly. Therefore, the expression pattern of Ham doesn't seem to be a good example of dynamic expression in INPs. Can the authors clearly indicate how many genes have dynamic expression patterns in INPs and what criteria, i.e. based on fold changes, is applied to rank and select Opa? Currently, it appeared to be hand-picked.

We have followed reviewer’s suggestions and now included a list of dynamic genes with the selection criteria in subsection” Transcriptome analysis of distinct INP temporal states”. Opa was chosen as it scored the highest fifth among the dynamically expressed genes (49 genes in total).

4) Is asymmetric division of INPs impaired and resulting in the change of INPs numbers upon loss-of-opa or opa overexpression?

To address this, we performed pH3, Miranda and aPKC staining in control, opa knock-down and opa overexpression brains. We have found in all cases, Miranda and aPKC localized to opposite poles as expected in asymmetric cell division, indicating that asymmetric cell division is not affected (Figure 2—figure supplement 2E and Figure 5—figure supplement 1D-E). We rather think that these observed differences in total INPs numbers are due to the known connections between temporal patterning and cell cycle (Ey was shown to be required to end INP proliferation). Upon opa loss, we believe that the prolonged early temporal patterning allows more cell divisions for each INP, resulting in an increased cumulative number of INPs. In the case of opa over-expression, where D is skipped and Ey comes earlier, we would expect the opposite phenomenon to happen.

5) We appreciate the excellent quality of the FACS sorting of the three temporally distinct INP populations and are very impressed with how cleanly the authors were able to isolate the 3 populations, and how well they have documented it! This makes the RNA-seq data particularly invaluable to the field.

We thank the reviewer for this comment.

6) With respect to Opa and its place in the temporal transitions, we are largely in agreement with the authors and their interpretations. They propose a model where Opa represses D and suggest that Opa also activates Grh. However, in their subsequent interpretations they seem to consider Opa's repression of D as its main mechanism of action in temporal progression. Would not Opa's activation of Grh, leading indirectly to repression of D, tie in all the data from this and the Byaraktar and Eroglu papers better? It would explain why D^+^, Opa^+^ double positive INPs are seen at all, why D-/- INPs can still progress through the temporal cascade, and why overexpressing Opa resulted in loss of D^+^ INPs and increase in GrH^+^, Ey^+^ INPs (incomplete expression of Grh in UAS-Opa shown in Figure 5 might be due to presence of Ham). Furthermore, it would not call upon differential levels of expression of Opa (which the authors have not shown at the protein level), as timing of expression/inhibition of these genes as the INP divides might account for the temporal transitions.

We thank the reviewer for this great suggestion. We indeed cannot completely exclude that Opa’s main role would be to activate Grh, leading to an indirect D repression. However, Bayraktar et al. showed that Grh LOF led to prolonged D expression, which was eventually repressed, while Grh GOF didn’t affect the number of D^+^ or Ey^+^ INPs. These data suggest that Grh is not sufficient to repress D and would argue against Opa acting solely on Grh activation.

7) If the authors agree with this, have they looked at the DM1 lineage, which is known to not express Grh? Is Opa expressed in this lineage? If not, does misexpression of Opa there result in Grh activation and D repression? The authors have analysed this in DM2 and DM3 – the same brains could be analysed for DM1.

We thank the reviewer for this great suggestion. Opa is expressed in all type II lineages, including DM1 (Figure 4—figure supplement 1E). We analyzed opa’s role in DM1 lineages which don’t express Grh naturally, with both loss and gain of function experiments. Our results showed that opa is required in DM1 lineages in order to repress D expression (Figure 2—figure supplement 1A-C). Furthermore, in the case of opa overexpression, D^+^ INP numbers were significantly decreased (Figure 5—figure supplement 2A-C). Nonetheless, opa overexpression didn’t cause ectopic Grh expression in DM1 lineages (Figure 5—figure supplement 2B). These data altogether further support that opa’s main function is to repress D expression.

8) The authors must have verified the Opa:V5 tool before using it. Could they please describe this?

We apologize for this oversight. We did verified the opa::V5 transgenics by Western blot (Figure 4 —figure supplement 1A) and by IF upon knock-down looking at V5 expression (Figure 4—figure supplement 1F-G).

*9) The authors show that D comes up in INPs before Opa does. However, this is not very clear from the images. Could the authors maybe show magnified insets of these types INPs with the two reporters co-localised? Am I right in understanding that apart from this one difference, Opa is co-localised with every other temporal factor in the INPs? Related to this, the authors find that Opa RNA levels are high early and late in the lineage, and dip in the middle. Do the protein levels reflect these dynamics? Their discussion seems to suggest it does.*

We have now included the close-up images of D and opa localization (Figure 4A). We further included opaV5 staining together with Grh and Ey (Figure 4B and Figure 4—figure supplement 3A). To analyze the differential expression level of opa in INPs, we measured the intensity of opaV5 levels in these three INP subpopulations (Figure 4A-D and Figure 4—figure supplement 3A-B). Indeed, we found that D^+^ INPs have higher opaV5 levels than D^-^ INPs, while GrH^+^ INPs have lower levels than Grh^-^ INPs. Finally, Ey^+^ INPs have similar levels of opaV5 levels as Ey^-^ INPs, which is expected as Ey^-^ INPs contain both opa^high^ D^+^ and opa^low^ GrH^+^ INPs.